# De novo active sites for resurrected Precambrian enzymes

Valeria A. Risso[1], Sergio Martinez-Rodriguez[1], Adela M. Candel[1], Dennis M. Krüger[2], David Pantoja-Uceda[3], Mariano Ortega-Muñoz[4], Francisco Santoyo-Gonzalez[4], Eric A. Gaucher[5], Shina C.L. Kamerlin[2], Marta Bruix[3], Jose A. Gavira[6] & Jose M. Sanchez-Ruiz[1]

Protein engineering studies often suggest the emergence of completely new enzyme functionalities to be highly improbable. However, enzymes likely catalysed many different reactions already in the last universal common ancestor. Mechanisms for the emergence of completely new active sites must therefore either plausibly exist or at least have existed at the primordial protein stage. Here, we use resurrected Precambrian proteins as scaffolds for protein engineering and demonstrate that a new active site can be generated through a single hydrophobic-to-ionizable amino acid replacement that generates a partially buried group with perturbed physico-chemical properties. We provide experimental and computational evidence that conformational flexibility can assist the emergence and subsequent evolution of new active sites by improving substrate and transition-state binding, through the sampling of many potentially productive conformations. Our results suggest a mechanism for the emergence of primordial enzymes and highlight the potential of ancestral reconstruction as a tool for protein engineering.

[1] Departamento de Quimica Fisica, Facultad de Ciencias University of Granada, 18071 Granada, Spain. [2] Science for Life Laboratory, Department of Cell and Molecular Biology, Uppsala University, BMC Box 596, S-751 24 Uppsala, Sweden. [3] Departamento de Quimica Fisica Biologica, Instituto de Quimica Fisica Rocasolano, CSIC, c/Serrano 119, 28006-Madrid, Spain. [4] Departamento de Quimica Organica, Facultad de Ciencias University of Granada, 18071 Granada, Spain. [5] School of Biology, School of Chemistry and Biochemistry, Parker H. Petit Institute for Bioengineering and Biosciences, Georgia Institute of Technology, Atlanta, Georgia 30322, USA. [6] Laboratorio de Estudios Cristalograficos, Instituto Andaluz de Ciencias de la Tierra, CSIC-University of Granada Avenida de la Palmeras 4, Granada, 18100 Armilla, Spain. Correspondence and requests for materials should be addressed to S.C.L.K. (email: kamerlin@icm.uu.se) or to J.M.S.-R. (email: sanchezr@ugr.es).

The generation of completely new active sites capable of enzyme catalysis is, arguably, one of the most fundamental unsolved problems in protein science. Rational design approaches to this problem have often used complex computational methods, have targeted simple model reactions and have typically led to low levels of catalysis[1]. These studies would seem to suggest, therefore, the unlikelihood of the emergence of completely new active sites in non-catalytic scaffolds. Certainly, most modern enzyme functions likely evolved from previously existing functionalities. On the other hand, most biochemical processes are extremely slow in the absence of enzymes[2] and specialized enzymes are likely to have catalysed many different reactions already in the last universal common ancestor[3,4]. It could be inferred from this that efficient mechanisms for the emergence and subsequent evolution of completely new enzyme functionalities must exist or, at least, that they must have existed at the primordial protein stage. However, little is known about such mechanisms.

Buried and partially buried ionizable groups with perturbed properties often play essential catalytic roles in modern enzymes. Single hydrophobic-to-ionizable residue mutations that generate partially buried groups with perturbed properties may have plausibly provided a feasible route to the generation of completely new active sites. This notion has been previously proposed[5,6] but never tested in practice. Conformational flexibility could have assisted the emergence of the new enzyme functionalities and its subsequent evolution by facilitating substrate and transition-state binding, through sampling a greater number of potentially productive conformations. Many years ago, Jensen proposed in a highly influential article[7] that primordial enzymes were capable of catalysing a diversity of reactions. It is conceivable that the conformational flexibility that is likely linked to such a broad generalist nature[8–10] may have facilitated the emergence of new enzyme functionalities in the first place.

Here, we explore and test these notions using resurrected Precambrian β-lactamases[11,12] as scaffolds for the engineering of completely new active sites. To date, only a handful of systems have been studied using ancestral resurrection, and only a few of these resurrection efforts targeted 'old' (~3 billion years) phylogenetic nodes (Fig. 2 of ref. 13). Of these systems, resurrected Precambrian β-lactamases have been thoroughly characterized in terms of their structure, function and stability[11]. These putative ancestral proteins have been shown to be highly stable and able to efficiently degrade several antibiotics[11]. Previous[9] and current computational analyses, as well as new NMR relaxation studies reported here, support that this substrate promiscuity is linked to enhanced conformational flexibility. The broad substrate scope of resurrected Precambrian β-lactamases may reflect the wide variety of substances these enzymes had to hydrolyse[11,12]. We do not claim, therefore, that they are necessarily at Jensen's ancestral generalist stage, although, strictly speaking, this possibility should not be ruled out. In any case, the available resurrected Precambrian β-lactamases[11] should provide an adequate model with which to address the role of ancestral conformational flexibility in the emergence of new enzyme functionalities. More generally, we have recently reviewed[14] several arguments and recent publications that support that promiscuity may be a common outcome of ancestral protein resurrection.

Here, we use carefully selected systems that span a vast region of the sequence space of both ancestral and modern β-lactamases, and probe the enzymatic features that allow for the emergence of a non-natural activity in these enzymes, as well as why it only appears in a specific snapshot of evolutionary time. We demonstrate the role of conformational flexibility in allowing for the emergence of new enzymatic functions, as well as its

importance in the subsequent evolvability of the enzyme. Finally, our data highlights the potential of ancestral reconstruction as a tool for protein engineering by providing far more powerful evolutionary starting points than can be obtained from modern enzymes.

## Results

**Selected model systems**. In the present study, we have used the proteins encoded by the most probabilistic sequences at six Precambrian phylogenetic nodes in the evolution of class A β-lactamases (Fig. 1). The reconstructed sequences and the procedure used to obtain them have been described in detail in Supporting Information of ref. 11. These proteins display large sequence differences between themselves (Supplementary Table 1) and they are properly folded, highly stable, active and share the β-lactamase fold[11]. In addition, we have also used here proteins encoded by alternative sequences at the GNCA node (common ancestor of Gram-negative bacteria). As is customary in the field, these alternative sequences were derived[11,13] from a random weighted sampling of the posterior probability distribution. They differ from the most probabilistic sequence at 8–20 positions (Supplementary Tables 2 and 3). Finally, for comparison, we have also used 10 modern β-lactamases (Fig. 1) that have been considered in the literature to be archetypical examples of β-lactamases and are well characterized in terms of their structure and function[15]. These modern proteins provide a fair representation of the β-lactamases from the several bacterial taxa (Fig. 1) and they span modern β-lactamase sequence space to a substantial extent, as they show limited sequence identity between themselves (Supplementary Table 4). Note also their limited sequence identity with the putative ancestral β-lactamases studied here (Supplementary Table 5). Overall, we explore in this work a vast region of the sequence space of both ancestral and modern β-lactamases.

We have selected the Kemp elimination reaction (Fig. 2) as our primary target for the reaction to be catalysed by the generated new active site, for several reasons. First, it provides a simple activated model for proton abstraction from carbon, which is a fundamental chemical process that underlies many biochemical reactions. Completely new active sites are expected to display low catalysis levels (high catalysis levels would be the outcome of subsequent evolution) and their emergence is best probed by activated substrates. Second, Kemp elimination is a non-natural reaction (unknown in biological organisms). No natural enzyme has evolved, therefore, to catalyse this reaction[16] and studies on engineered Kemp eliminases are unlikely to be compromised by contamination from natural enzymes. Following from this, Kemp elimination has been often used as a benchmark for rational enzyme design[1]. Therefore, use of the same system allows for a quantitative comparison of the catalytic efficiencies of the designed constructs presented in this work to those generated in previous enzyme design studies.

**Generation of a new active site for Kemp elimination**. The possibility that slow conformational changes (on the microseconds to seconds timescale) play roles in enzyme catalytic cycles has been proposed and explored[17]. Consequently, we used NMR relaxation determinations (see Methods section and Supplementary Methods) on the β-lactamase encoded by the most probabilistic sequence at the GNCA node (GNCA$_{MP}$ β-lactamase) to guide our design of a new active site. This putative ancestral protein displays a large number of residues with conformational contribution to the relaxation rates determined by NMR (see the comparison with the modern TEM-1 β-lactamase shown in Fig. 3). A large accumulation of

such residues is visually apparent in the region encompassing helix h1 (residues 26–41), helix h11 (residues 271–290) and the loops 225–229 and 252–257. Residue 229 within this region appears as a suitable target for the generation of a new active site through a hydrophobic-to-ionizable residue replacement. A substantially buried and highly conserved tryptophan residue is present at position 229 in both modern and reconstructed Precambrian β-lactamases. The indole side chain of tryptophan has a shape similar to that of the Kemp substrate. Precisely because of this shape congruence, replacement of W229 to a new residue with basic properties will not generate a new active site

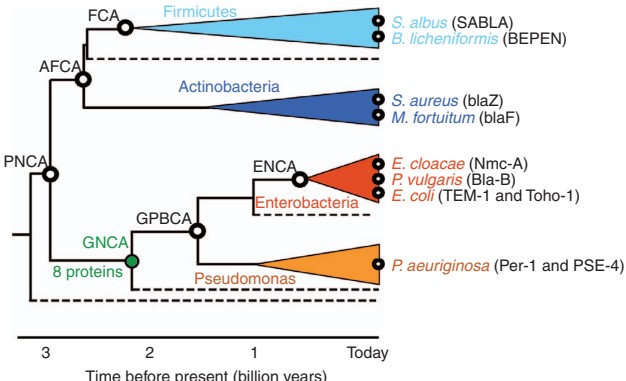

**Figure 1 | Ancestral and modern β-lactamases used as scaffolds in this work.** Schematic representation of the phylogenetic tree for class A β-lactamases[11]. The ancestral nodes studied in this work correspond to the common ancestors (CA) of Firmicutes (FCA), Actinobacteria and Firmicutes (AFCA), Enterobacteria (ENCA), Gammaproteobacteria (GPBCA), various Gram-negative bacteria (GNCA) and various Gram-positive and Gram-negative bacteria (PNCA). The proteins encoded by the most probabilistic sequences at these nodes[11] were prepared and used as scaffolds for engineering. In addition, seven alternative reconstructions at the GNCA node (Supplementary Tables 2 and 3) were also prepared; these are labelled GNCA1 to GNCA7, with the most probabilistic sequence at the node being labelled GNCA_MP. β-lactamases from 10 modern organisms are also studied in this work and these organisms are shown at the right.

capable of catalysing the Kemp elimination unless, of course, conformational rearrangements occur in the enzyme to be able to avoid steric clashes between the substrate and the new residue.

We found that a simple W229D replacement leads to substantial levels of Kemp elimination activity in the β-lactamases at all ancestral nodes studied here (Supplementary Table 6), with the only exception of the evolutionary recent ENCA node (Fig. 1). By substantial activity, we mean that the observed levels of Kemp elimination activity were clearly distinguishable (and much higher in most cases) than the background (enzyme-free) levels (Supplementary Fig. 1). In contrast, the W229D variants of all 10 modern β-lactamases studied led to levels of Kemp elimination activity that, even at protein concentrations of about 20 μM, could be barely distinguished from the background levels (Supplementary Fig. 1). It is important to note that the W229D variants of all 10 modern β-lactamases studied did show antibiotic degradation activity (linked to the natural active site) at nM concentrations (Supplementary Fig. 2). Therefore, their lack of Kemp elimination activity cannot be attributed to the 'disruptive' W229D mutation preventing their folding, which is anyhow a problem that is less likely to arise with the highly stable ancestral β-lactamases. It is worth noting here that enhanced stability is a common outcome of Precambrian protein resurrection[11,18–20], likely linked to the thermophilic nature of early life.

A wide diversity of experimental results confirm (or are consistent with) the main features of the designed approach used. Specifically, X-ray crystallography in the presence of 5(6)-nitrobenzotriazole, a known transition-state analogue of the Kemp elimination reaction (Fig. 4) and inhibition by this analogue (Supplementary Fig. 3) support that Kemp elimination does occur at the site generated by the W229D replacement. The catalytic role of the aspartate at position 229 is further confirmed by mutational studies: the W229G variant shows negligible activity and the nearby aspartate at position 228 does not have a catalytic role, as the replacement of D228 with A does not substantially impair the activity (Supplementary Table 6). The catalytic role of the aspartate at position 229 is also confirmed by the pH dependence of the catalysis, which is consistent with the raised pK value expected for an aspartate residue in a hydrophobic environment (Fig. 5). Following from this, 3D-structure determination in the presence of

**Figure 2 | Reactions and compounds studied in this work.** (**a**) The Kemp elimination of 5-nitrobenzioxazole (5-nitro-benzo[d]isoxazole). A schematic transition state structure is shown here; (**b**) tryptophan; (**c**) 5(6)-nitrobenzotriazole (a transition-sate analogue) and (**d**) indole (the tryptophan side chain). Finally, (**e**) the hydrolysis of *p*-nitrophenyl acetate is also shown here.

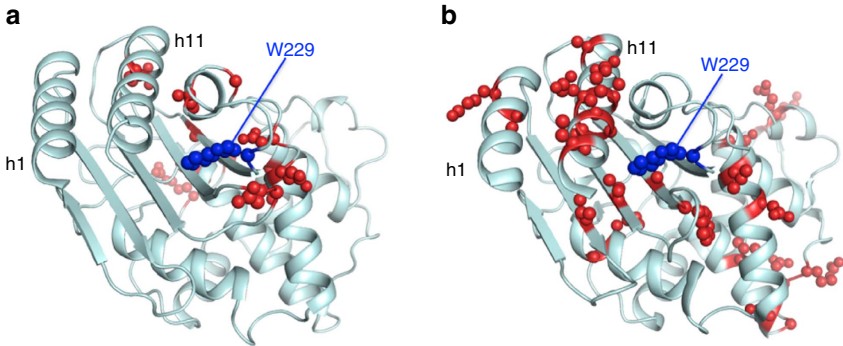

**Figure 3 | NMR relaxation studies on modern and ancestral β-lactamases.** The 3D-structures of (**a**) TEM-1 β-lactamase (PDB 1BTL) and (**b**) the β-lactamase encoded by the most probabilistic sequence at the GNCA node[11] (PDB 4B88) are displayed. The residues for which the relaxation rates cannot be explained without including a conformational exchange contribution (ref. 31 and this work) are highlighted in red. The h1 and h11 α-helices are labelled. The tryptophan residue at position 229 is highlighted in blue.

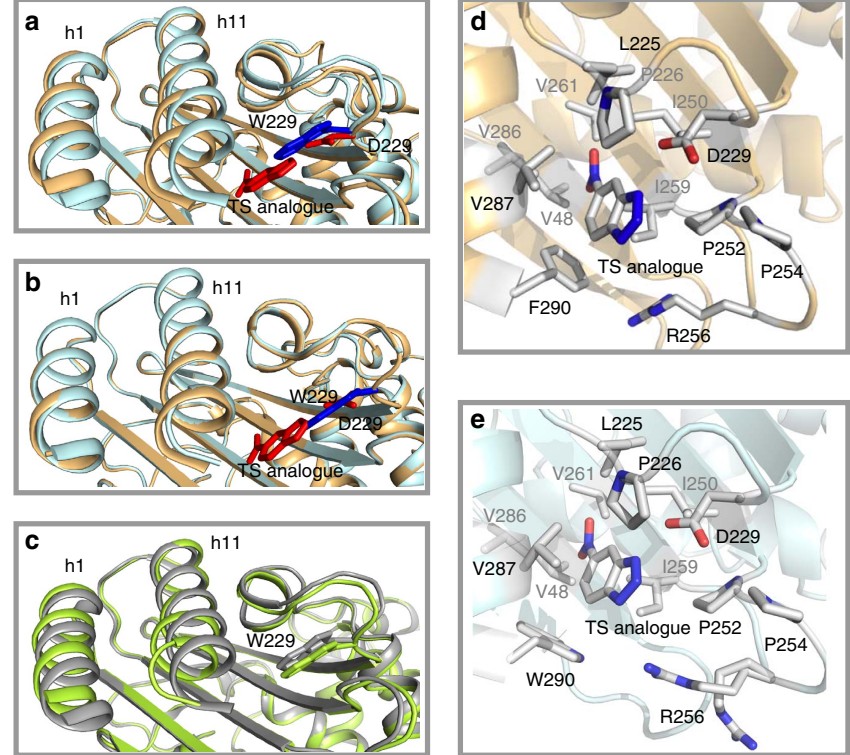

**Figure 4 | 3D-structures of the *de novo* Kemp eliminases.** (**a,b**) Shown here are the structures of the W229D variant of GNCA$_{MP}$ β-lactamase and the W229D/F290W variant of GNCA**4** β-lactamase, respectively. In both cases, the structures with 5(6)-nitrobenzotriazole, a transition-state analogue, bound at the *de novo* active site (orange-coloured structures with analogue in red) are superimposed with the 3D-structures of the corresponding GNCA$_{MP}$ and GNCA4 backgrounds (light-blue coloured structures with W229 in darker blue). The shift of the h1 and h11 α-helices is apparent in **a**, but not in **b**. The reason for this is that the h1 and h11 helices are already displaced in the background GNCA**4** variant, as shown by the superposition of the two backgrounds in **c** (GNCA$_{MP}$ is shown in grey and GNCA**4** is shown in light green). (**d,e**) blow-ups of the new active site region in the structures of the W229D variant of GNCA$_{MP}$ β-lactamase (**d**) and the W229D/F290W variant of GNCA**4** β-lactamase (**e**) with the bound transition-state analogue.

the transition-state analogue (Fig. 4) confirms the role of conformational flexibility in the generation of a new function, as transition-state binding is shown to rely on conformational rearrangements. Specifically, the bound transition-state analogue is displaced with respect to the position originally occupied by the tryptophan 229, as required by the presence of an aspartate residue at position 229 in the active variants. Such a displacement is made possible by a shift of the h11 α-helix (and the concomitant shift of the substantially solvent-exposed h1 α-helix). We note also that the catalytic efficiency for the Kemp elimination in the engineered ancestral proteins correlates with the transition-state analogue binding constants derived from

inhibition experiments (Supplementary Fig. 4), supporting that catalysis is linked to transition-state stabilization. Finally, catalysis of Kemp elimination is enhanced by an additional F290W amino replacement in the neighbourhood of position 229, which likely stabilizes the transition state through a face-to-edge interaction with the new tryptophan residue (Fig. 4).

**Comparison with previous rational designs.** Kemp elimination is a non-natural reaction that is unknown to biological organisms. No enzyme is, therefore, expected to have evolved to catalyse this

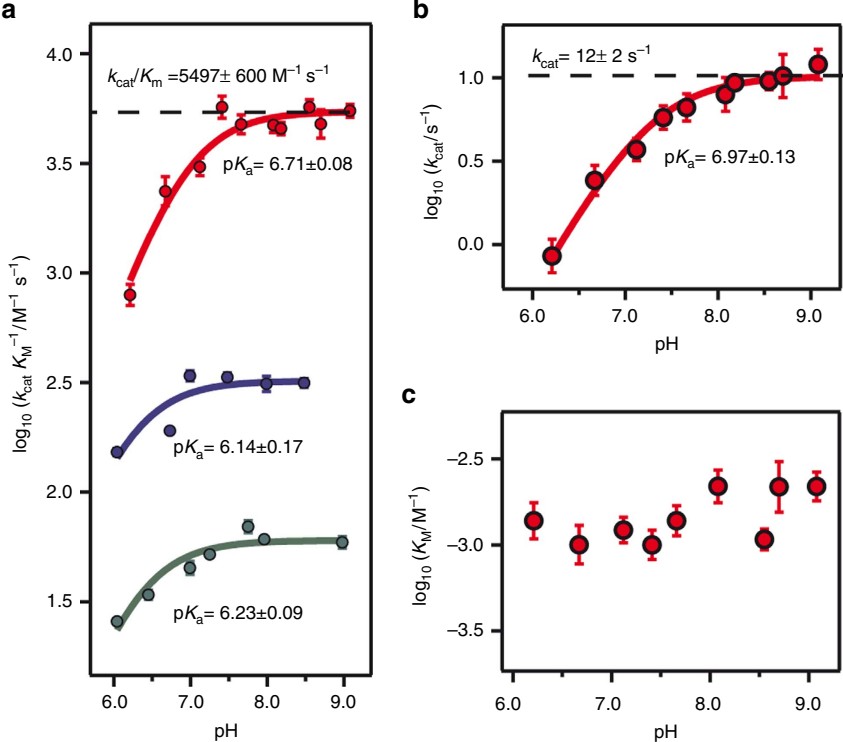

**Figure 5 | pH dependence of the *de novo* Kemp eliminases.** (**a**) Plot of catalytic efficiency ($k_{cat}/K_M$) versus pH. (**b**) Plot of turnover number ($k_{cat}$) versus pH. (**c**) Plot of the Michaelis constant ($K_M$) versus pH. The data shown are derived from the fitting of the Michaelis–Menten equation to the experimental profiles of rate versus substrate concentration. Error bars stand for the s.d.'s of the fitting parameters as provided by the fitting program used (Igor Pro 6.37). The colour of the data points refers to the β-lactamase variant studied, specifically: GNCA$_{MP}$-W229D (green); GNCA**3**-W229D (blue); GNCA**4**-W229D/F290W (red). The continuous lines in (**a**,**b**) represent the best fits of equation (1) to the experimental data. The p$K$ values determined from such fits are shown alongside the lines, together with the standard errors derived from the fittings. The high pH limiting values for our best Kemp eliminase, GNCA**4**-W229D/F290W, are shown as dashed lines. $K_M$ values for this variant (shown in **c**) appear to be essentially pH-independent.

reaction and, consequently, Kemp elimination activity is not expected among natural proteins. Indeed, Hilvert and coworkers[21] found no detectable Kemp eliminase activity in calmodulin, barnase, lysozyme, trypsin, chymotrypsinogen and chymotrypsin. Furthermore, Tawfik and coworkers[16] screened the ASKA library (∼4,300 clones) for natural and non-natural activities and found only two enzymes that could catalyze Kemp elimination, although these appeared to be promiscuous activities. On the other hand, serum albumins can catalyse Kemp elimination[22] with a turnover number that is about $10^{-2}\,s^{-1}$ at neutral pH, but that approaches $10\,s^{-1}$ at pH ∼10. However, serum albumins, even in the absence of bound metals, can catalyse a remarkable diversity of non-natural reactions[23], due to hydrophobic pockets, which can bind different substrates and to the presence of lysine residues with catalytic properties within those pockets.

The capability of albumins to catalyse the Kemp elimination reaction (among several other non-natural reactions) suggests that the scarcity of natural enzyme catalysts for such a simple reaction is simply due to the lack of a selective pressure to generate Kemp elimination activity during evolution. On the other hand, rational design efforts to engineer artificial Kemp eliminases have resulted in limited success[1], despite the simplicity of the targeted reaction. Indeed, comparison with these previous Kemp eliminase designs (Fig. 6) provides further evidence that the catalytic efficiencies ($k_{cat}/K_M$ values) we have generated in the ancestral β-lactamases are indeed substantial and consequential. The lower end of the variation range spanned by the W229D and W229D/F290W variants of the 12 successful ancestral backgrounds used is actually similar or clearly above the results previously reported using

minimalist design[24,25]. The upper end of the range is above the results previously reported using more complex design approaches[26,27] (iterative design and design based on Rosetta), which also involved large numbers of mutations to reach those efficiencies, and it is less than two orders of magnitude below the catalytic efficiency for the best Kemp eliminase reported to date[28] (which is the outcome of 17 rounds of directed evolution from an iterative design background).

Regarding the turnover numbers, the $k_{cat}$ values obtained for the W229D/F290W variants of the ancestral β-lactamase scaffolds are up to approximately seven orders of magnitude above the rate of the uncatalysed reaction (Fig. 6). For comparison, the best artificial Kemp eliminase reported to date[28] displayed a approximately nine orders of magnitude enhancement, but, as noted above, this was obtained as a result of 17 rounds of directed evolution on a designed background with already substantial activity. Furthermore, the upper end of the range spanned by our $k_{cat}$ values is clearly above the reported $k_{cat}$ values for complex designs that required much larger numbers of mutations (Fig. 6). Michaelis plots for the single W229D variants are linear and do not allow $k_{cat}$ values to be calculated. However, lower limit estimates of $k_{cat}$ can indeed be derived from suitable analysis of linear Michaelis plots (see Supplementary Fig. 5 for details). These estimated lower limit values for the single W229D variants are shown with open circles in Fig. 6. Remarkably, they are similar to the actual $k_{cat}$ determined for the double W229D/F290W variants. Therefore, the $k_{cat}$ enhancement of up to seven orders of magnitude over the uncatalysed reaction is actually produced by the single W229D mutation, while F290W mostly favors substrate and transition-state binding.

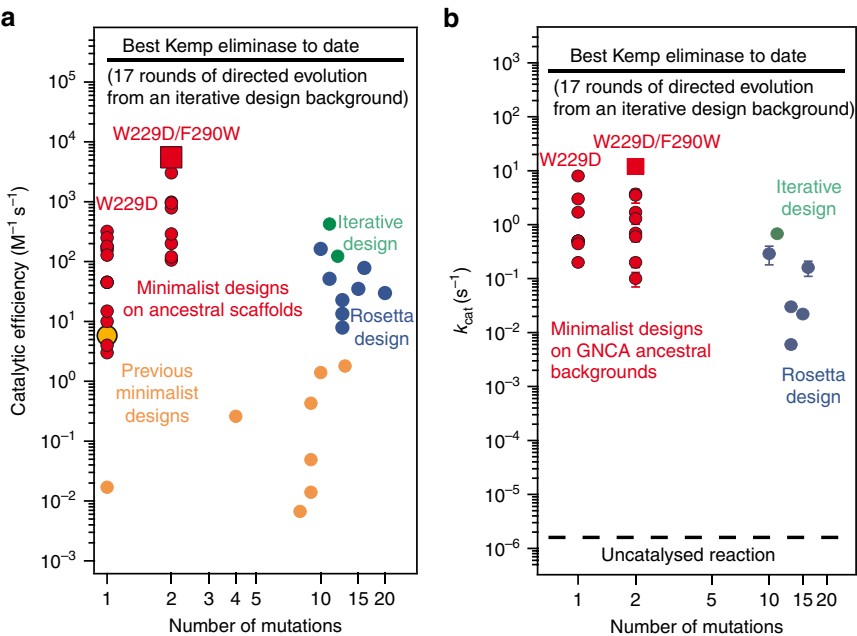

**Figure 6 | Comparing our Kemp eliminases with previous rational designs. (a)** The catalytic efficiencies ($k_{cat}/K_M$ values) of the single (W229D) and double (W229D/F290W) mutant variants of the ancestral β-lactamases studied here are shown in red. The red square data point represents the high pH value for GNCA4-W229D/F290W (Fig. 4a). The values for the 'previous minimalist designs' are taken from Korendovych et al.[24] (large orange data point at number of mutations value of unity) and Merski and Shoichet[25] (small orange data points). In both cases, the design is based on a single mutation, but, in the latter study[25], additional mutations had to be introduced for mainly stabilization purposes (the values shown correspond to the variants displayed in Fig. 3 of Merski and Shoichet[25]). The values obtained using a design approach that involved Rosetta are taken from Röthlisberger et al.[26]. Only the values for the eight designs that led to substantial Kemp eliminase activity are shown (59 designs were actually tested by Röthlisberger et al.[26]). Iterative design values are taken from Privett et al.[27]. Here, 'iterative' means that an original design with very low activity was improved on the basis of 3D-structural information and molecular dynamics simulations to achieve substantial levels of catalysis. The efficiency of the best Kemp eliminase reported to date[28] is also shown here for comparison. **(b)** Comparison of the Kemp eliminase turnover numbers ($k_{cat}$ values) obtained in this work with those reported for previous rational designs. The meaning of the symbols is the same as in **a**. The values shown as open symbols for the W229D variants of the ancestral backgrounds are actually lower limit estimates calculated as described in Supplementary Fig. 5. We also show here the reaction rates corresponding to the best artificial Kemp eliminase reported to date[28] and to the uncatalysed reaction for comparison.

Finally, it must be noted that catalysis of Kemp elimination by carboxylic acids is strongly accelerated in aprotic solvents. The acetate ion in acetonitrile is in fact an excellent catalyst of the Kemp elimination with a reported second order rate constant of $2,800\,M^{-1}\,s^{-1}$, a value that has been used as a metric to judge the catalytic efficiency of artificial Kemp eliminases[1]. Unlike previous rationally designed Kemp eliminases (Fig. 6a), our best eliminase displays a maximum catalytic efficiency ($\sim 5,500\,M^{-1}\,s^{-1}$, Fig. 5a) that exceeds the acetate in acetonitrile level.

**Additional activities of the engineered ancestral enzymes.** We used our best eliminase (the W229D/F290W variant of the alternative GNCA4 reconstruction at the GNCA node), to test whether the introduction of the new active site affects the catalysis at the 'old' natural active site (catalytic residue S70, located at about 23 Å from position 229). β-Lactamases encoded by reconstructed sequences corresponding to the GNCA node have been previously shown[11] to be able to degrade a variety of antibiotics, including penicillin and third-generation antibiotics with efficiencies similar to that of a modern average enzyme (by contrast, the modern TEM-1 β-lactamase is a specialist enzyme that displays high catalytic efficiency with penicillin and a substantially lower efficiency with third-generation antibiotics). We therefore determined the Michaelis–Menten parameters for the degradation of a penicillin antibiotic (benzylpenicillin) and one third-generation antibiotic (cefotaxime) catalysed by GNCA4-W229D/F290W (Fig. 7). We found levels of catalysis

for antibiotic degradation similar to those previously reported for the GNCA_MP β-lactamase (Supplementary Table 7).

We also used our best eliminase to explore the potential promiscuity at the generated new active site. Our results show that the GNCA4-W229D/F290W β-lactamase does in fact also catalyse the hydrolysis of p-nitrophenyl acetate (Figs 2 and 7 and Supplementary Fig. 6), a substrate commonly used to assess esterase activity[16]. However, lactam hydrolysis and ester hydrolysis are chemically similar and the natural (antibiotic degradation) active site could therefore in principle contribute to the observed esterase activity. Nevertheless, our results (Fig. 7) indicate a minor natural-site contribution that only becomes apparent when the de novo active site is saturated at the higher substrate concentrations. This is specifically shown by the fact that the esterase activity of GNCA4-W229D/F290W is inhibited by 5(6)-nitrobenzotriazole, a transition-state analogue of the Kemp elimination reaction (Supplementary Fig. 3). In addition, replacing the catalytic serine at the antibiotic degradation active site with alanine does not significantly impair the rate of p-nitrophenyl acetate hydrolysis (Fig. 7), except at the higher substrate concentrations at which the de novo active site is saturated. Similarly, blocking the antibiotic degradation active site through the irreversible reaction with clavulanic acid does not significantly impair the rate of p-nitrophenyl acetate hydrolysis (Fig. 7), except at the higher substrate concentrations at which the de novo active site is saturated. Finally, saturating concentrations of benzylpenicillin do not significantly impair the rate of p-nitrophenyl acetate hydrolysis (Supplementary Fig. 7), except

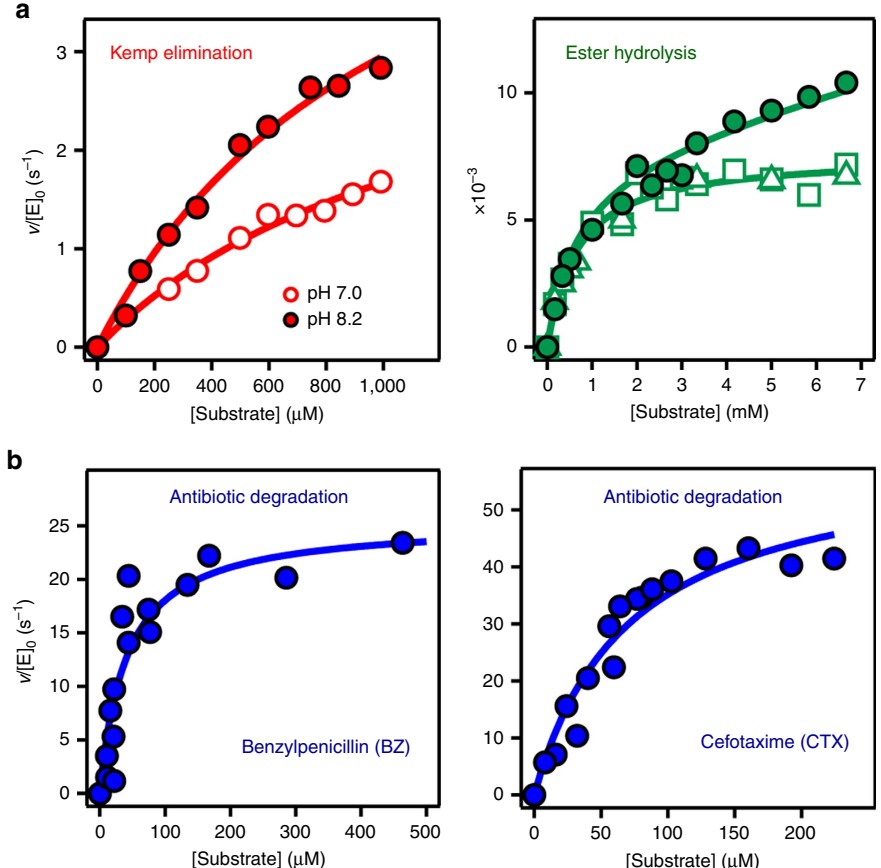

**Figure 7 | Various activities of the W229D/F290W variant of the GNCA4 scaffold.** (**a**) Activities linked to the new active site. (**b**) Activities linked to the natural preexisting active site. Plots of rate versus substrate concentration are shown in (**a,b**), with the continuous lines representing the best fits based on the Michaelis–Menten equation. Ester hydrolysis data correspond to *p*-nitrophenyl acetate hydrolysis and were obtained with the GNCA**4**-W229D/F290W variant (closed data points) and also with this variant modified to block any esterase activity at the natural, antibiotic degradation site. This was achieved by mutating the catalytic S70 to alanine (open squares) or by inactivation by clavulanic acid (open triangles). Catalytic parameters for ester hydrolysis discussed in the Main Text are derived from the Michaelis–Menten fit to the modified variants. The equation used to fit to the data of the unmodified variant includes an additional linear term to account for esterase activity at the antibiotic degradation site. The esterase catalytic efficiency at the natural active site is, however, found to be ∼20-fold smaller than that at the new active site.

at the higher substrate concentrations at which the *de novo* active site is saturated.

The catalytic efficiency for the hydrolysis of *p*-nitrophenyl acetate at the new active site, $k_{cat}/K_M = 11.7 \pm 1.1 \, M^{-1} \, s^{-1}$, is similar to the best values reported in the literature for rationally designed esterases[29]. Furthermore, the turnover number at the new active site, $k_{cat} = 7.6 \times 10^{-3} \pm 2 \times 10^{-4} \, s^{-1}$, indicates a three orders of magnitude enhancement over the rate of the non-enzymatic reaction[30] ($k_{uncat} = 2.5 \times 10^{-6} \, s^{-1}$; Supplementary Fig. 6). These are remarkable results, in particular since, we are dealing here with a new active site generated on the basis of a minimalist approach that did not target esterase activity.

**Computational modelling of engineered ancestral enzymes.** To further explore and confirm the role of conformational flexibility in the emergence of new enzyme functions, we have complemented our experimental work with molecular dynamics (MD) simulations of the wild-type and mutant forms of the modern TEM-1 and *Bacillus licheniformis* (BL) β-lactamases, as well as the ancestral ENCA, GNCA$_{MP}$, GNCA**4** and PNCA β-lactamases (Fig. 1). We chose these particular proteins both due to the availability of X-ray structures (see the Methods section) and because they span the range of Kemp elimination activities obtained in this work upon new active site generation. In

particular, the modern TEM-1 and BL β-lactamases, as well as the comparatively recent ENCA β-lactamase (Fig. 1), display negligible activity upon the W229 mutation, while this mutation confers substantial Kemp elimination activity to the PNCA, GNCA$_{MP}$ and GNCA backgrounds following the order PNCA < GNCA$_{MP}$ < GNCA**4**. Furthermore, the double W229/F290W variant of the GNCA4 scaffold is the most active Kemp eliminase reported in this work with a rate enhancement of about seven orders of magnitude over the rate of the uncatalysed reaction. As described below, our MD simulations support that conformational flexibility contributes substantially to these activity trends.

We have used the root mean square fluctuation (RMSF) of each amino acid in our simulations as a measure of the overall flexibility of the system. We examined first the GNCA$_{MP}$ and TEM-1 β-lactamases, that is, the two proteins for which NMR relaxation data are available (Fig. 3) from this and previous[31] work, respectively. As explained in the second section of the Results, the region of the ancestral scaffold structure targeted for new active site generation is characterized by a large number of residues with conformational exchange contributions to the relaxation rates. Clearly, complete agreement between the results of the MD simulations and the conformational exchange contributions to the NMR relaxation rates is not to be

expected, mainly because of the different timescales involved. That is, inclusion of exchange terms to explain relaxation rates reveals dynamic processes in the micro to milliseconds range[32], while shorter timescales are typically probed by MD simulations[33]. Nevertheless, we find a clear correspondence between the calculated $C_\alpha$ RMSF values and the number of residues with conformational exchange contribution to NMR relaxation at the region of the new active site (Supplementary Fig. 8). It is therefore likely that the approximately microsecond conformational transitions at the region of the new active site in the ancestral scaffold are reflected in our 200 ns ($= 0.2\,\mu s$) MD simulations, which actually approach the lower end of the conformational exchange timescale range.

The capability of our MD simulations to capture flexibility features that are relevant for new active site generation is further supported by the general congruence with the catalytic properties of several modern and ancestral scaffolds. In the case of the wild-type enzymes, a clear increase can be seen in the flexibility of residues 252 through to the h11 helix upon moving from TEM-1 and BL to the GNCA$_{MP}$ to GNCA4 β-lactamases, which are the residues that are primarily involved in forming the cavity creating the *de novo* active site (Fig. 8). The flexibility of this region appears indeed to follow the general activity trend described above. Also, the largest overall changes in flexibility can be seen in helices h1 and h11, as well as the loops covering the *de novo* active site, in agreement with both the crystallographic data (Fig. 4) as well as with the NMR relaxation data shown in Fig. 3. The mobility of different regions of these enzymes has also been highlighted in Fig. 9. The function-generating mutation W229D appears to increase the flexibility, which is the likely outcome of the introduction of a charged residue in a hydrophobic environment[6]. Still, the effect is much more pronounced in the ancestral β-lactamases, while the modern β-lactamases show only a comparatively small increase in flexibility (Fig. 8).

We subsequently performed MD simulations in the presence of the transition-state analogue 5(6)-nitrobenzotriazole (TSA) to explore the impact of transition-state binding on the conformational dynamics, as well as how well each system could accommodate the transition state in its new active site. The observed TSA binding features do correlate with the activity

trends and with the flexibility trends defined by the RMSF data. The W229D variants of the ancestral scaffolds that give rise to substantial Kemp elimination activity (PNCA, GNCA$_{MP}$ and GNCA4) can accept the TSA in the MD simulations, although the TSA dissociates from the less active W229D variant of the PNCA scaffold after about 50 ns (Fig. 10). By contrast, efficient TSA binding is not observed with the W229D variants that show very low Kemp elimination activity. Specifically, we could not obtain a stable TSA complex with the W229D variant of ENCA β-lactamase, while the TSA flies out of the cavity in the W229D variants of the modern TEM-1 and BL β-lactamases within about 20 ns (Fig. 10). We have also observed the same pattern (low flexibility in the region of the *de novo* active site and inability to retain the TSA bound) for the β-lactamases from *Enterobacter cloacae* (Nmc-A) and *Proteus vulgaris* (Bla-B) (Supplementary Fig. 9). This pattern, therefore, is likely to be a general feature of modern β-lactamases from Enterobacteria (Fig. 1).

Finally, there are two apparent exceptions to the flexibility/activity trend discussed above. First, ENCA β-lactamase develops substantial flexibility in the new active site region upon the W229D mutation, even though this protein shows negligible Kemp elimination activity. However, D229 is partially deprotonated at neutral pH and flexibility upon introduction of a charged group in a hydrophobic environment reflects, to some extent, the conformational fluctuations required to allow some water penetration and the consequent stabilization of the partially buried charge[6]. Such a specific kind of flexibility might not be relevant to the understanding of transition-state binding. Indeed, there should not be significant water penetration when the transition state is bound to the new active site, and the negative charge of the catalytic aspartate is actually stabilized by the interaction with the proton being abstracted. Second, the GNCA4 scaffold appears to be more rigid (in the new active site region) than the GNCA$_{MP}$ scaffold, despite the fact that GNCA4 leads to higher levels of Kemp elimination activity upon the W229D mutation. However, 3D-structure determination shows that the GNCA4 is actually more preorganized for transition-state binding than the GNCA$_{MP}$ scaffold (Fig. 4). This can again ultimately be viewed as a reflection of flexibility at the GNCA

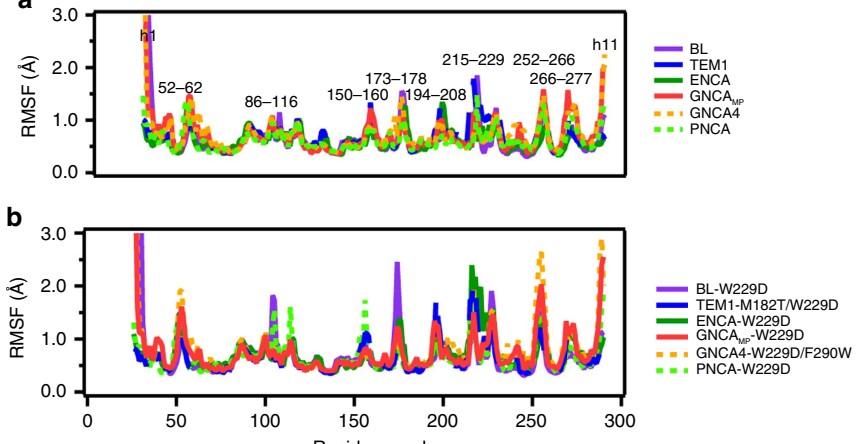

**Figure 8 | $C_\alpha$-atom root mean square fluctuations of all simulated variants.** Shown here are the root mean square fluctuations (RMSF, Å) of all $C_\alpha$ atoms during our simulations of (**a**) wild-type and (**b**) variant forms of the TEM-1 (blue), *Bacillus licheniformis* (purple), GNCA$_{MP}$ (red), GNCA4 (orange), ENCA (dark green) and PNCA (light green) β-lactamases. In the case of the variant forms, the simulations were performed both in the absence of the transition-state analogue (TSA) or in complex with the TSA, 5(6)-nitrobenzotriazole. The simulations shown in (**b**) were all performed in the absence of the TSA, and a corresponding comparison of the $C_\alpha$ RMSF values of the ancestral enzymes both with and without the TSA present are shown in Supplementary Fig. 29. All values shown here are averages over the last 60 ns of three independent trajectories (180 ns total simulation time), which were obtained as described in the Methods section.

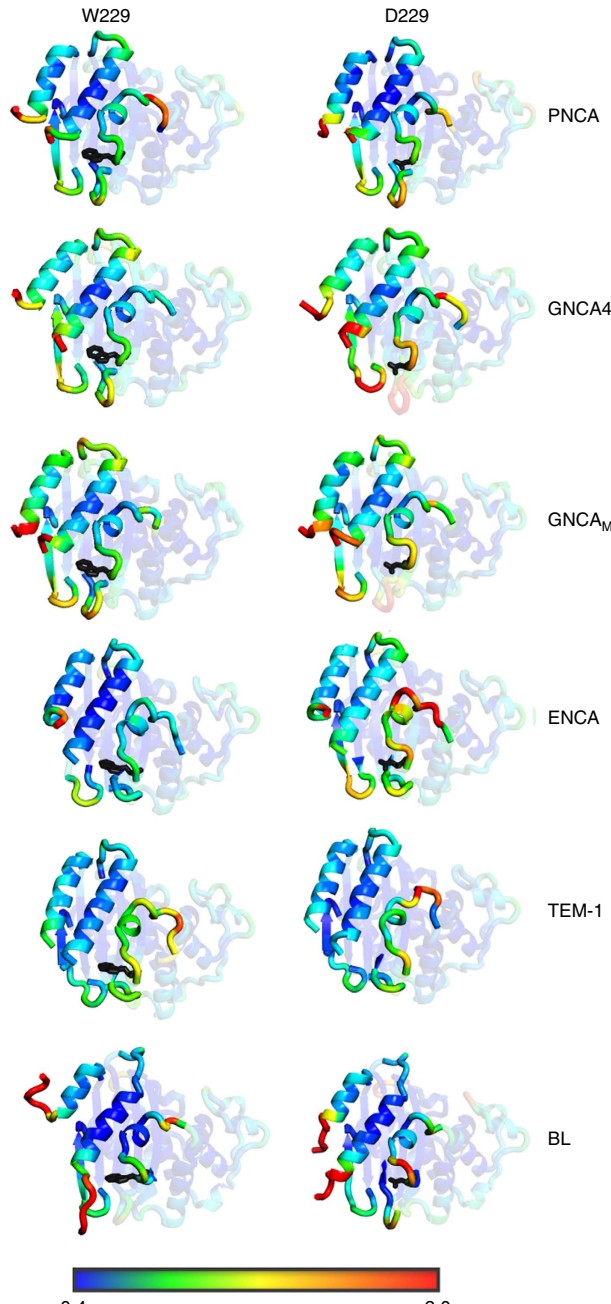

**Figure 9 | Tertiary structures of key β-lactamases coloured by RMSF.**
Shown here are the tertiary structures of the (left) wild-type and (right) variant forms of the TEM-1, ENCA, *Bacillus licheniformis*, GNCA$_{MP}$, GNCA4 and PNCA β-lactamases, coloured by the calculated root mean square fluctuations (RMSF) of their C$_\alpha$ atoms, based on the values shown in Fig. 8. The colour scale at the bottom of this figure shows the calculated RMSF, in Å. This figure is presented to allow for a visual comparison of the relative mobilities of the different enzymes relative to each other, and how this maps on to their tertiary structures. Note here that the *Bacillus licheniformis* β-lactamase has unstructured C- and N-terminal segments. The structures shown here are the most representative structures obtained from our molecular dynamics simulations, obtained from clustering analysis as described in the Methods section.

node, as mutations removed from the active site can easily shift the conformational equilibria at this node. Clearly, the specific mutations present in the GNCA**4** scaffold have led to the population of a conformation in the W229D variant with a cavity

where the substrate can easily form a reactive conformation, and with better architecture for effective transition-state stabilization.

## Discussion

Modern proteins can perform an enormous diversity of molecular tasks, often with high efficiency and specificity. Most of these modern functions evolved from previously existing functionalities. Yet, it is inescapable to assume that the emergence of completely new functions has also occurred, at least at some early stages in protein evolution. For instance, it has been recently estimated from the number of unique enzymes and the number of domain superfamilies[34] that at least 87% of all enzyme functions have evolved from another function or from ancestors with generic functionalities. This would leave a fraction of 13% of enzyme functions as plausible candidates for having emerged through the generation of new active sites. While the 13% figure may be an overestimation, as some cases of homology between superfamilies may be missed by sequence comparison[35], it does however suggest that the generation of new active sites may not actually be an exceedingly rare event and thus should garner more serious consideration. This is supported by reports of alternate-site promiscuous enzymes that display catalysis of a secondary reaction at a site other than the active site of the natural catalytic process[36]. Furthermore, most fundamental biochemical processes are extremely slow in the absence of enzymes[2] and seemingly specialized enzymes are likely to have catalysed many different reactions already in the last universal common ancestor[3,4]. It may be reasonably inferred that efficient mechanisms for the emergence and subsequent evolution of new enzyme functionalities must exist or, at least, that they must have existed at the primordial protein stage. Little is known, however about these mechanisms. Recent work[29] has shown that introducing a catalytic group in the hydrophobic cavity of the C-terminal domain of calmodulin can generate significant levels of catalysis for simple reactions. The simple mechanism for the emergence of new enzyme functions demonstrated here does not require the recruitment of a preexisting hydrophobic cavity. A hydrophobic-to-ionizable amino acid replacement generates a buried (or partially buried) residue with the perturbed properties that are useful in catalysis and conformational flexibility assists the generation of a completely new active site by facilitating substrate and transition-state binding. We have shown that, when using resurrected Precambrian β-lactamases as scaffolds for protein engineering, a minimalist design approach based on this mechanism leads to levels of catalysis for the Kemp elimination reaction of up to approximately seven orders of magnitude above the rate of the uncatalysed reaction, as well as to significant ester hydrolysis activity. The role played by conformational flexibility in the generation of the new function is clearly apparent in the X-ray crystallographic structures of our designed enzymes and supported by the extensive computational simulations reported, as well as by experimental binding studies (Supplementary Fig. 10).

The process of inferring ancestral sequences inherently generates uncertainly. It is therefore customary in the field to test phenotypic robustness by studying not only the protein encoded by the most probabilistic sequence at a given node, but also the proteins encoded by alternative reconstructions derived from random sampling from the posterior probability distribution at the node. We have previously used this approach to demonstrate phenotypic robustness for the antibiotic degradation activity at the natural active site of the GNCA β-lactamase[11]. Remarkably, however, the new active site behaves differently and engineered β-lactamases encoded by several alternative reconstructions at the GNCA node (common ancestor of

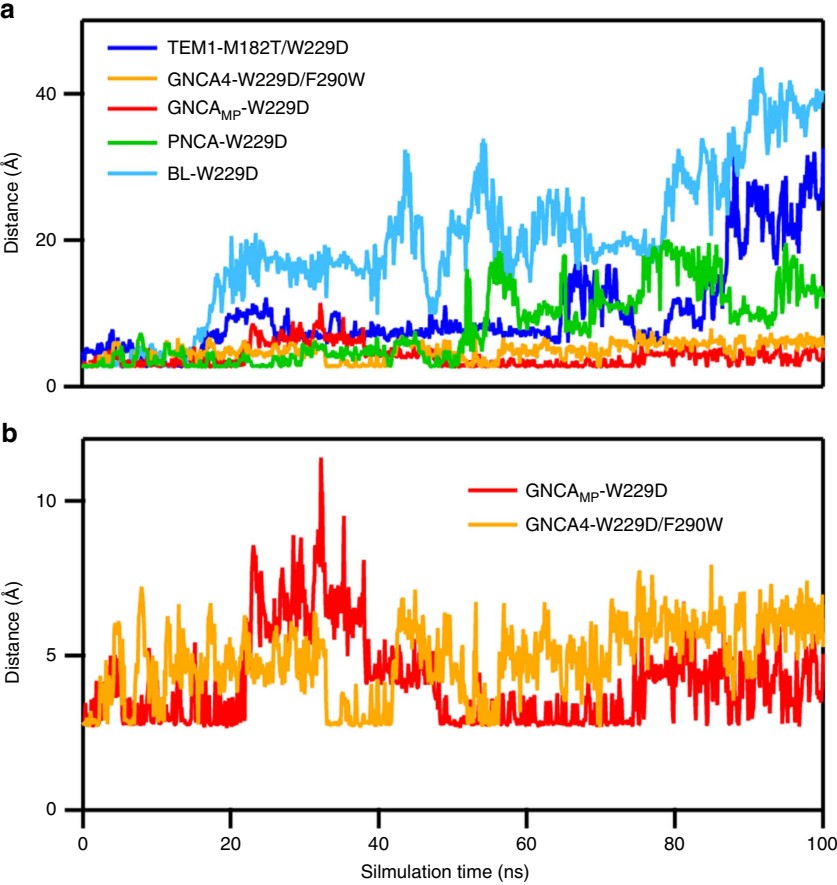

**Figure 10 | Distance monitoring during our simulations of selected β-lactamases.** Monitoring the time evolution of the distance (in Å) between the nitrogen atom of the 5(6)-nitrobenzotriazole transition-state analogue (TSA) and the closest oxygen atom of D229, in simulations of enzyme-TSA complexes of the TEM-1 (blue), *Bacillus licheniformis* (purple), GNCA$_{MP}$ (red), GNCA4 (orange) and PNCA (green) β-lactamase variants. Shown here are (**a**) a comparison of calculated distances for all systems, and (**b**) a close-up for the ancestral β-lactamases. Note that in **b**, the large values obtained for the GNCA$_{MP}$ β-lactamase variant between ∼20 and 40 ns of simulation time are due to flipping of the TSA in the binding pocket, and not due to dissociation from the active site as is the case for the modern β-lactamases and the PNCA β-lactamase shown in **a**. All values shown here are averages over three independent trajectories, which were obtained as described in the Methods section.

Gram-negative bacteria) display Kemp elimination activities that span a ∼30-fold range. The several GNCA sequences studied differ among themselves at 8–20 positions, which are, in essentially all cases, distal from the neighbourhood of the Kemp elimination active site (Supplementary Table 3). Therefore, the ∼30-fold range of catalytic efficiency observed suggests long-range communication between residues, a feature that is often linked to conformational dynamics[37–39]. It emerges that function at new active sites may be more sensitive to long-range effects than function at 'old' highly evolved active sites. In any case, the existence of such long-range effects makes it difficult to make a meaningful comparison between the different ancestral scaffolds in terms of their capability to generate a new function and no such detailed comparison is attempted here. From a more general viewpoint, however, it is clear that long-range effects could contribute to the immediate evolvability of newly emerged functions as they imply that many mutations are able to modulate the new activities. Accordingly, conformational dynamics/flexibility may contribute to the emergence of new functions as well as to their subsequent evolution. Certainly, reaching high levels of catalysis for increasingly complex reactions will likely require the concomitant evolution, sophistication and specialization of the catalytic machinery at the active site. Still, conformational flexibility may have played a key role even at the early specialization stages, as the concomitant

long-range mutational effects may have helped avoid stasis between the improvements in the catalytic machinery caused by rare mutations at the active site.

Finally, it is remarkable that the minimalist approach that generates substantial levels of a new function in most of the ancestral β-lactamases tested here only leads to catalysis levels barely distinguishable from the background when performed on 10 modern β-lactamases (Supplementary Fig. 1). This different behaviour is plausibly linked to differences in conformational flexibility between the ancestral and the modern proteins, as our attempts to generate a substantial Kemp elimination activity in the modern β-lactamases by removing potential steric interferences to transition-state binding were unsuccessful (Supplementary Note). Therefore, the distinct ancestral versus modern pattern of new function generation found appears to provide direct evidence in support of recent proposals about the potential of resurrected ancestral proteins as scaffolds for protein engineering[11,40].

## Methods

**Protein expression and purification.** Purification of the different β-lactamases studied in this work followed the procedures we have previously described[11] with minor modifications. Briefly, the genes were cloned into a pET24 vector with kanamycin resistance and transformed into *Escherichia coli* BL21(DE3) cells. Ni-NTA affinity chromatography was used to purify the His-tagged proteins used in most of the experiments reported in this work. We checked that the presence of a

His-tag does not substantially impair catalysis (Supplementary Table 6 and Supplementary Fig. 11).

$^{15}$N-labelled and $^{13}$C,$^{15}$N-labelled proteins for NMR experiments were purified (without His-tag) from cells grown in M9 minimal medium containing either $^{15}$NH$_4$Cl or $^{15}$NH$_4$Cl and ($^{13}$C$_6$)D-glucose (Cambridge Isotopes Laboratories) as the sole sources of nitrogen and carbon, respectively. Mutations were introduced using the quickchange lighting system site-directed mutagenesis kit (Agilent) and were checked by DNA sequencing.

**Activity assays.** Kemp elimination activity assays were performed at 25 °C in HEPES 10 mM, 10 mM sodium phosphate or 10 mM borate (in all cases with 100 mM NaCl), depending on the pH range. We found no significant difference between the rates determined in these buffer systems at overlapping pH values. Product formation was followed spectrometrically at 380 nm and rates were calculated using an extinction coefficient of 15,800 M$^{-1}$ cm$^{-1}$. All activity measurements were corrected by a blank performed under the same conditions. Our experimental protocols for the determination of Kemp elimination activity reproduce literature data on the catalysis of this reaction by serum albumins and acetate ion in acetonitrile/water (Supplementary Figs 12 and 13). The pH dependence of the catalysis was fitted using the following equation:

$$rate = \frac{A \cdot 10^{(pH - pK)}}{1 + 10^{(pH - pK)}} \quad (1)$$

where 'rate' stands for the catalytic efficiency or for the rate at a given substrate concentration. $A$ and $pK$ are fitting parameters. Equation (1) assumes that the pH dependence of catalysis is determined by a single $pK$. A simple analysis based on the Michaelis–Menten mechanism shows that the $pK$ in the free enzyme is probed by the catalytic efficiency versus pH profile, while the $pK$ in the Michaelis–Menten complex is probed by the catalytic efficiency versus pH profile.

Degradation of the antibiotic nitrocefin was assayed at 25 °C in HEPES buffer 10 mM, 100 mM NaCl, pH 7.0 and followed spectrophotometrically at 486 nm. A change of extinction coefficient of 17,400 M$^{-1}$ cm$^{-1}$ was used to calculate degradation rates from the time dependence of the absorbance. Degradation of benzylpenicillin and cefotaxime was assayed as we have previously described in detail[11].

Inhibition of the antibitotic degradation activity by clavulanic acid, sulbactam and tazobactam was tested by incubation with 1 mM inhibitor at 4 °C in HEPES buffer 25 mM, pH 7.0. We found the complex of the W229D/F290W variant of GNCA4 β-lactamase with clavulanic acid to be very stable (but not the complexes with sulbactam and tazobactam), as shown by the fact that the nictrocefin degradation activity was essentially eliminated even after overnight incubation. Specifically, nitrocefin degradation levels after overnight incubation at 4 °C were less than 0.5% those of a non-inhibited sample. For this reason, clavulanic acid was used (after overnight incubation) in the experiment aimed at eliminating the esterase activity linked to the natural/ancestral active site (Fig. 7a).

*p*-Nitrophenyl acetate hydrolysis was followed spectrophotometrically, essentially as described by Moroz et al.[29]. We did not observe, however, the burst phase linked to the formation of an acyl intermediate.

**Protein crystallization and structure determination.** Crystals of GNCA$_{MP}$ ('WT' and W229D mutant, 28 mg ml$^{-1}$) and GNCA4 ('WT' and the W229D/F290W mutant, 25 mg ml$^{-1}$) β-lactamases were grown in capillaries by the counter-diffusion technique[41] using 5 M sodium formate, 0.1 M sodium acetate pH 4.0 (GNCA$_{MP}$-W229D), 2 M NH$_4$SO$_4$ 0.1 M Tris-HCl pH 8.0 (GNCA$_{MP}$) or a PEG (polyethylene glycol) mixture (PEG 400/4000/8000; 20, 10 and 15%, respectively), Tris-HCl 0.1M pH 7.0 (GNCA4 and GNCA4-W229D/F290W) as precipitating agents. In the case of the unliganded forms, crystals were extracted from the capillary and equilibrated with the mother liquid supplemented with 15% (v/v) glycerol before flash-cooling them in liquid nitrogen. Co-crystallization of the TS-analogue-liganded forms was unsuccessful. Therefore, the liganded forms were obtained by immersion of a portion of capillary containing crystals of GNCA$_{MP}$-W229D or GNCA4-W229D/F290W into a solution containing the original precipitant composition but at pH 9.0, supplemented with 1 mM of the transition-state analogue (5)6-nitrobenzotriazole and 15% (v/v) glycerol. The crystals were equilibrated for 48 h before being extracted from the capillaries and flash-cooled. Data were collected at beam lines ID30A, ID29 and ID23-1 (ESRF) and XALOC (ALBA). The procedures used for 3D-structure determination, as well as the data collection and refinement statistics, are summarized in Supplementary Table 8 see also Supplementary Figs 14–19 for stereo plots of relevant portions of the electronic density maps and Supplementary Fig. 20 for a structural comparison between the modern and ancestral β-lactamases in the region of the *de novo* active site. The resulting coordinates and structure factors have been deposited at the PDB with the accession codes 5FQI, 5FQQ, 4UHU, 5FQJ, 5FQK and 5FQM.

**NMR spectroscopy.** We provide here a brief summary of the NMR studies on the GNCA$_{MP}$ β-lactamase. For a more detailed account, Supplementary Methods, Supplementary Figs 21 and 22 and Supplementary Tables 9–11. NMR experiments were performed at 31.5 °C on a Bruker AV 800 spectrometer equipped with a cryoprobe on a 0.6 mM uniformly $^{13}$C,$^{15}$N-labelled sample. Sequence-specific

assignments were made using standard procedures with the following experiments: 2D $^{1}$H–$^{15}$N HSQC and 3D HNCO, HN(CA)CO, HN(CO)CA, HNCAi, CBCA(CO)NH and HNCACB. Data obtained with these experiments were complemented with those of specific amino acid type discrimination. The only unassigned residues in GNCA$_{MP}$ β-lactamase were Ala26, Ala27, Ser70 and Thr237.

$^{15}$N relaxation parameters T$_1$, T$_1$ρ, T$_2$ and {$^1$H}-$^{15}$N NOE were acquired on a Bruker AV 600 spectrometer equipped with a cryoprobe, at 31.5 °C on a 0.6 mM, buffered pH 6.7, uniformly $^{15}$N-labelled sample, following standard procedures. Relaxation times were calculated by least-squares fitting of peak intensities to a two-parameter exponential function. Heteronuclear NOEs were calculated from the ratio of cross-peak intensities in spectra collected with and without amide proton saturation during the recycle delay. Uncertainties in peak heights were determined from the standard deviation of the distribution of intensities in the region of the HSQC spectra where no signal was present and only noise was observed.

The principal components of the inertia tensor were calculated with the PDBinertia program[42] using the X-ray structure of the GNCA$_{MP}$ β-lactamase (PDB ID: 4B88). We estimated the overall correlation time from the ratio of the mean T$_1$ and T$_2$ values. These mean values of T$_1$, T$_1$ρ and T$_2$ were calculated from a subset of residues with little internal motion and no significant exchange broadening. The diffusion tensor, which describes rotational diffusion anisotropy, was determined by standard approaches and the $^{15}$N relaxation was analysed assuming dipolar coupling with the directly attached proton (with a bond length of 1.02 Å), and a contribution from the $^{15}$N chemical shift anisotropy evaluated as  − 172 p.p.m. Residues with conformational exchange contribution were determined from the fits of several extensions of the Lipari and Szabo model to the experimental relaxation data[43,44]. We ascertained that application of the analysis approach described above reproduced the NMR relaxation parameters published for TEM-1 β-lactamase[31]. This allows us to make meaningful comparisons between the dynamic features of the ancestral protein (reported here) and those for the modern protein[31]. Both proteins appear very rigid on the ps-ns timescale with an order parameter of $S^2 \sim 0.9$ for all residues in both cases. There is, however, a clear difference on the timescale (μs-ms) commonly associated with biological processes such as substrate binding, allosteric regulation and catalytic cycle[17]. Specifically, the number of residues for which the relaxation rates cannot be explained without including a conformational exchange contribution ($R_{ex}$) is 26 for GNCA$_{MP}$ β-lactamase versus 12 for TEM-1 β-lactamase (Fig. 3).

**Synthesis of 5(6)-nitrobenzotriazole.** This compound was obtained following the reaction conditions reported by Wasik et al.[45]. Briefly, a solution of NaNO$_2$ (230 mg, 3.3 mmol) in 1.5 ml of water was added to a mixture of 4-nitro-*o*-phenylenediamine (306 mg, 2 mmol) in 4 ml of AcOH and 1.5 ml of water at 0°C. The solid was collected by filtration and washed with H$_2$O to afford 300 mg (91% yield) of pure light yellow solid that was dried under vacuum over P$_2$O$_5$; mp 213–214 °C (lit. 215–216 °C). 1D-NMR spectrum of the compound is given in Supplementary Fig. 23.

**Spectrophotometric monitoring of ligand binding.** Binding of the transition-state analogue to variant β-lactamases was followed on the basis of the inhibition of the Kemp eliminase activity or esterase activity of this protein. Briefly, stock solutions of the transition-state analogue and indole in acetonitrile (approximate concentrations 0.1 M and 5 M, respectively) were prepared and different microliter amounts were added to 2 ml solutions of the β-lactamase. After 1 h incubation, activity was determined as described above. Longer incubation times led to essentially identically results. The final acetonitrile concentration in the assayed solution was always below 2%.

Dissociation constants for inhibitors were determined from the fitting to the experimental data of an equation based on the Michaelis–Menten mechanism with reversible competitive inhibition:

$$rate = \frac{A}{1 + [S]/K_M + [I]/K_I} \quad (2)$$

where $[S]$ and $[I]$ are the substrate and inhibitor concentrations, $A$ is a constant and $K_I$ is the inhibitor dissociation constant. For W229D variants of the GNCA scaffolds, plots of rate versus substrate concentration are linear (Supplementary Fig. 24), implying that $[S]/K_M$ is much smaller than unity and can be neglected in the denominator of equation (2). For the W229D/F290W variants, the $K_M$ in equation (2) was fixed in the value obtained from the Michaelis–Menten analysis of the rate versus substrate concentration profiles (Supplementary Fig. 25).

**Molecular dynamics simulations.** To complement the experimental data, we also performed MD simulations on the wild-type *B. licheniformis*, *E. cloacea* (NMC-A), *P. vulgaris* (Bla-B), TEM-1, ENCA, GNCA$_{MP}$, GNCA4 and PNCA β-lactamases, as well as the corresponding GNCA$_{MP}$-W229D, GNCA4-W229D/F290W and PNCA-W229D variants (PDB IDs: 4BLM, 1BTL, 3ZDJ, 4B88, 5FQQ, 4C6Y, 4UHU, 5FQI respectively). We also performed simulations on the BL-W2299D, NMC-A-W229D, BLA-B-W229D, TEM-1-M182T/W229D and PNCA-W229D variants; however, as crystal structures of three enzymes were unavailable, we manually inserted these substitutions into the wild-type enzyme using PyMOL's Mutagenesis

Wizard[46]. The specific GNCA variants were selected on the basis of their observed Kemp eliminase activities (Supplementary Table 6), and the variant simulations were performed both in complex with the transition-state analogue (TSA) 5(6)-nitrobenzotriazole (PDB IDs for the β-lactamase-TSA complexes: 5FQJ for GNCA$_{MP}$-W229D, 5FQK for GNCA4-W229D/F290W), as well as in the TSA free form. For all other variants, the TSA was manually placed in the newly created cavity in agreement with structural alignment of the crystal structures from the TSA bound ancestral variants. All MD simulations were performed with the GPU implementation of the AMBER16 simulation package[47], and the ff14SB force field[48,49]. The Sander module was used for the initial minimization and equilibration runs, and PMEMD for the production runs in explicit solvent[50,51]. To obtain ff14SB compatible parameters for the TSA, we parameterized this compound using ANTECHAMBER[52] and Gaussian09 (ref. 53). Partial charges were obtained using the standard RESP fitting procedure[54,55], and the ligand atoms were described by the GAFF force field[48]. All non-standard parameters generated by the GAFF force field are provided in the Supplementary Table 12.

All model systems were generated with LEaP. Each model system was placed in an octahedral TIP3P water box[56,57], which allowed the box to extend to at least 11 Å from the solute in each direction. The protonation state of all ionizable residues was determined by empirical p$K_a$ calculations using PROPKA 3.1 (ref. 58), and by visual inspection (in particular in the case of histidine side chains). On the basis of this, ionizable residues were all kept in their standard protonation state at physiological pH, and all histidine residues were kept neutral. This led to total system charges of $-7$, $-5$, $-7$ and $-8$ for TEM-1, BL, GNCA4 and GNCA$_{MP}$ respectively. These systems were then neutralized by addition of the appropriate number of Na$^+$ counterions depending on the total charge of the system. The solvated systems were then subjected to a two-step minimization procedure to remove clashes between the water molecules and the solute. This was comprising of 50 steps of steepest descent and 200 steps of conjugate gradient minimization using 25 kcal mol$^{-1}$ Å$^{-2}$ harmonic positional restraints on all solute atoms, followed by 50 steps of steepest descent and 200 steps of conjugate gradient minimization with weaker 5 kcal mol$^{-1}$ Å$^{-2}$ harmonic positional restraints on all solute atoms. After this initial minimization, the following four sequential equilibration steps were performed: a 50 ps NVT simulation to increase the thermostat target temperature using the Berendsen thermostat and pressure control algorithms[59] with 0.5 ps time constants for both the bath coupling and pressure relaxation; a 50 ps NPT simulation at a constant isotropic pressure of 1 atm to adjust the density of the system to 1 g cm$^{-3}$; five 50 ps NVT simulations in which remaining 5 kcal mol$^{-1}$ Å$^{-2}$ harmonic positional restraints were progressively decreased in 1 kcal mol$^{-1}$ Å$^{-2}$ increments, allowing us to finally perform a 50 ps NVT simulation without any restraints on the system.

For each system, we performed three independent equilibrations using different initial random seeds, the starting points for which were obtained by increasing the target thermostat temperature from 100 to 299.9 K, 300.0 and 300.1 K, respectively, to obtain three independent simulations. This was followed by performing 100 ns production simulations for each of the three replicas. All simulations were performed using a 2 fs time step, saving snapshots every 20 ps. The SHAKE algorithm[60,61] was used during the dynamics to constrain all bonds involving hydrogen atoms. Short-range non-bonded interactions were calculated subject to an 8 Å cut-off radius. Long-range interactions were described using the particle mesh Ewald method[62,63]. The temperature was kept constant at 300 K using Berendsen's weak coupling algorithm[59]. All production simulations were performed under NVT conditions. Subsequent analyses of the MD trajectories were performed using CPPTRAJ[64]. The first 40 ns of each simulation was discarded from the analysis as equilibration time, and all RMSF and clustering data shown in the paper are obtained as averages over the last 60 ns of three independent equilibration runs for each system (that is, 180 ns total simulation time per system). The backbone r.m.s.d. plots shown in Supplementary Fig. 26 demonstrate that all systems remained stable over the remaining simulation time. In Supplementary Figs 27 and 28, we compare the average RMSF values per residue with the corresponding averages of the B-factors derived from X-ray crystallography. Although B-factors have been often interpreted in dynamical terms[65], they are affected by a large number of other factors, including the resolution limit, radiation damage, crystal lattice defects, rigid body motions, occupancy levels and refinement artifacts[66,67]. For the protein systems studied here, we observe an approximate congruence between the B-factors and RMSF values that appears comparable to the congruence reported for other protein systems in the literature[68].

**Data availability.** X-ray crystallographic data for the several 3D-structures determined in this work have been deposited at the PDB with the accession codes 5FQI, 5FQQ, 4UHU, 5FQJ, 5FQK and 5FQM. Force field files for the transition-state analogue, as well as sample simulation input files and PDBs have been uploaded to Dryad, and can be accessed using the following DOI: 10.5061/dryad.53629. Additional data supporting the findings of this study are available within the article and its Supplementary Information files, and from the corresponding authors upon reasonable request.

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

## Acknowledgements

This work was supported by Feder Funds, Grants from the Spanish Ministry of Economy and Competitiveness BIO2015-66426-R (J.M.S.-R.), CSD2009-00088 (J.M.S.-R.), CTQ2011-29299-C02-01 (F.S.-G.), CTQ2011-22514 (M.B.), BIO2016-74875-P (J.A.G.), 'Factoría Española de Cristalizacioñ', Consolider-Ingenio 2010 (J.A.G.) and CEI BioTic V19-2015 (V.A.R.), a Wallenberg Academy Fellowship (S.C.L.K.) and DuPont Young Professor Award (E.A.G.) and Grants NNX13AI08G and NNX13AI10G (E.A.G.) from NASA Exobiology. The European Research Council has provided financial support under the European Community's Seventh Framework Programme (FP7/2007–2013)/ERC Grant Agreement No. 306474. We acknowledge the ESRF and ALBA for provision of synchrotron radiation time at beam lines ID29, ID23-1 and ID30A-1, and Xaloc, respectively, and the staff for their helpful support. Finally, we are grateful to the Swedish National Infrastructure for Computing (SNIC, 2015/16–12) for their generous provision of computational resources.

## Author contributions

V.A.R. obtained the protein variants, and designed, performed and analysed the experiments addressed at determining their natural and non-natural activities. S.M.-R. crystallized protein variants and determined their X-ray structure under the supervision of J.A.G, who also provided essential input regarding the interpretation of the 3D-structures. A.M.C. and D.P.-U. performed and analysed NMR relaxation experiments under the supervision of M.B. who also provided essential input regarding the interpretation of the relaxation data. D.M.K. performed the MD simulations under the supervision of S.C.L.K, who also provided essential input regarding the interpretation of the MD simulations. M.O.-M. synthesized the transition-state analogue under the supervision of F.S.-G, who also provided essential input regarding the properties of the transition-state analogue. E.A.G. provided methodology for bioinformatics analyses, as well as essential input for the interpretation of the results obtained in an evolutionary context. J.M.S.-R. and V.A.R. designed the research, with the exception of the computational work that was designed by S.C.L.K, J.M.S.-R., V.A.R. and S.C.L.K. wrote the manuscript. All authors discussed the manuscript and suggested modifications and improvements.

## Additional information

**Competing interests:** The authors declare no competing financial interests.

**Publisher's note**: 

