## [Peer Review File · Nature Communications]

Reviewer #1 (Remarks to the Author):

This is an interesting paper that examines secondary catalytic activity (Kemp elimination) in ancient sequences of beta lactamases. The authors do a thorough job of characterization of the resulting enzymes. Overall the work is of high quality although some modifications/clarifications are necessary.

1) The intro could be better written to justify the use of BL versus other enzymes. It is not clear to me what the rationale is.

2) It is not clear if the authors have a xray structure for all the enzymes analyzed by MD simulations. If they do not the discussion of subtle differences between enzyme flexibility and function seems premature. This should be clarified.

3) The Rex values are quite small ($< 2 \text{ s}^{-1}$). Given the data was only acquired at a single field there are alternative explanations for such small values that do not include us-ms motions. This should be discussed. Likewise, I echo the need for a structure for these analyses. It wasn't immediately obvious from the manuscript.

4) minor: DSC is mentioned in the Methods but I couldn't find any figure or results that included DSC.

Reviewer #2 (Remarks to the Author):

The findings of this study are fascinating and important. The results should definitely be published, and are suitable for Nat Comm. The authors' main discovery that a minimalist change (single amino acid substitution W229D) on ancestral (reconstructed pre-Cambrian) lactamase sequences can lead to significant Kemp elimination activities higher than those achieved by detailed designs (iterative, Rosetta) was unexpected and opens up many new avenues of research

into the evolution of protein function and structure.

There is room for improvement in the study and presentation of this work, however. It would be desirable if the following issues are addressed:

1. In Fig.6, catalytic efficiencies of the present minimalist designed Kemp eliminases are lower by about 2 orders of magnitude than the "best" Kemp eliminase to date from 17 rounds of directed evolution. How feasible is it to use the authors' minimalist sequences as basis for a directed evolution study to ascertain whether the final product of such an evolutionary process can exceed the best Kemp eliminases to date?

2. Experimental atomic structural data were provided for one ancestral sequence in Fig.3 (through NMR) and two ancestral sequences in Fig.4 (through X-ray crystallography), whereas MD simulation data were provided in Figs.8-10 for more sequences. This raises two questions:

(a) Conformational flexibility predicted by MD simulation should be compared directly with experimental structural data, e.g., between B-factors and the stringency of NMR constraints on one hand and the simulated RMSD in Fig.8 on the other hand. (b) Simulations should also be performed for the extant sequences (those shown on the right of Fig.1) as controls.

Reviewer #3 (Remarks to the Author):

There are several strong points to the manuscript. The combination of MD-simulations, experimental measurements (k_M , k_{cat} , NMR), and x-ray crystallography was compelling. Despite these strengths, there are some points which need further investigation prior for a publication in Nature communication:

1) First of all, I was wondering, how the authors could predict the sequences of the precambrian ancestors so precisely? I couldn't find anything about their sequence analysis in the methods part or supporting information.

2) Additionally to 1) the authors should implement a short paper research about the basic principle, that ancestral proteins are more promiscuous as modern ones.

3) The authors investigated whether the His-tag influences their reaction conditions or not. There is also a Figure of their results in the supplementary. I was wondering, however, why they did not measure k_{cat}/k_M , k_{cat} , and k_M in the absence of the His tag at pH-levels >8.0 . They did so for the protein with His-tag. Could the authors explain, why this is missing?

4) In the discussion of the paper the authors state that "87% of all enzyme functions have evolved from ancestors with generic functionalities and that 13% of enzyme functions have emerged through the generation of new active sites." Based on this I would expect that the modern lactamases (TEM-1, SABL, blaF, etc.) would show a stronger Kemp elimination than their ancestors. According to the results presented in this paper, however, this is not true, since the ancestors show a higher Kemp elimination rate than the modern lactamases. If it worked with the ancestors, why should the younger lactamases lose the ability for the Kemp elimination? Is this just because they are not so flexible anymore as their ancestors? If the modern lactamases lost the ability to perform a Kemp elimination than this is, in my opinion, more interesting than that the W229D/F290W alternative of GNCA4 ancestor has a higher Kemp elimination rate than the younger lactamases or the uncatalysed reaction.

5) The key aspects for their comparison of the activity of the ancestors and modern lactamases are based on two "model reactions": Kemp elimination and the hydrolysis of an ester, 4-nitrophenyl acetate. These reactions are characterized by a high thermodynamic driving force, low enthalpy of activation, and a single rate-determining transition state (Casey et al., J. Org. Chem., 1973 and Hollfelder et al., letters to Nature, 1996). I would ask the authors to repeat their experiments of the Kemp elimination with BSA and additionally in acetonitrile.

It seems churlish to ask the authors to do another experiment when they have already a lot, but without this crucial experiment, the paper remains a bit of a muddle, and all conclusions tendered must be heavily qualified and couched.

There were some smaller points:

- 1) In the caption of Figure 4 the picture 4E is not mentioned.
- 2) On p. 8, l. 224: the authors mentioned something about their ““best” eliminase”. Why is best written like “best”? This implies that it is not the best one.
- 3) On the same page in line 236 there is a t missing after throughthrough the irreversible...
- 4) On p. 10, l. 309 change varinant to variant
- 5) On p. 12 l. 349: I appears to me that an “an” is missing: ... reconstruction is an unavoidably ...

In summary, I found a lot here to support, but I also thought the paper was missing a) an explanation of the sequence analysis and b) a key experiment the comparison of the Kemp elimination with BSA and the repetition of this reaction in acetonitrile as solvent. Including both would make the paper much stronger. Including both it would certainly merit publication in Nature communication. Without it, it's a hard call—the paper becomes fundamentally descriptive, notwithstanding all the apparent quantification.

Reviewer #4 (Remarks to the Author):

The data presented, based on different biochemical approaches, do support the conclusions made by the authors on the relationship between conformational flexibility and directed evolution of enzymes.

I have only one minor comment on the manuscript. Can the author write "beta-lactamase or β -lactamase" instead of "lactamase"?

On the other hands, I have two major comments:

- 1) Page 8, lines 234-238 and figure 7.

The authors indicates that the replacement of the active site serine by alanine or the inactivation of the beta-lactamase by clavulanic acid do not affect the esterase activity. The figure 7 indicate that the absence of the β -lactamase activity decreases the esterase activity at high substrate concentration. Can the authors discuss this point?

In addition, although I am convinced that I could not find the procedure in the manuscript, we have no indication on the experimental procedure used to produce the clavulanic acid- enzyme complex.

In the case of the class A beta-lactamases, it is clear that clavulanic acid is a potent inactivator. Nevertheless, for different enzymes such as some TEM's variants or the *Bacillus licheniformis* beta-lactamase, the complex is unstable and can lead to its hydrolysis into free enzyme. Did the author check for the stability of the enzyme-clavulanic complex by mass spectrometry?

I would also suggest to the authors to perform a study of the impact of the presence of cefotaxime or benzylpenicillin on the esterase activity.

2) It is believed that active-site serine β -lactamases are the product of divergent evolution path of the transpeptidase domain of Penicillin Binding Protein. It has been also shown that active site serine β -lactamases can catalyze the hydrolysis of thioester substrate .

I would suggest to the authors to test their best esterase (GNCA4-W229D/F290W) for its activity against thioester or D-Ala-DAla

Reviewer #1 (Remarks to the Author):

This is an interesting paper that examines secondary catalytic activity (Kemp elimination) in ancient sequences of beta lactamases. The authors do a thorough job of characterization of the resulting enzymes. Overall the work is of high quality although some modifications/clarifications are necessary.

We thank the Reviewer for their positive assessment of the manuscript, and for their constructive comments, which we have addressed point-by-point below.

1) The intro could be better written to justify the use of BL versus other enzymes. It is not clear to me what the rationale is.

To date, only a few protein systems have been studied in the laboratory on the basis of ancestral resurrection (see Figure 2 of Gumulya and Gillan (2017)¹), and even fewer of these resurrection efforts have been able to reach very old (~3 billion years) phylogenetic nodes. Therefore, there are as yet not so many available options if one wants to use resurrected Precambrian proteins as scaffolds for engineering. We specifically selected resurrected Precambrian β -lactamases for the present work as they are one of the few systems that have been thoroughly characterized in terms of their structure, function and stability (in fact by one of the groups authoring the current manuscript). In addition, these enzymes display highly enhanced stability, which is a desirable feature in scaffolds to be used for protein engineering and directed evolution.

This issue has now been discussed in the revised version in a short paragraph (highlighted in red) on pages 3-4.

2) It is not clear if the authors have a xray structure for all the enzymes analyzed by MD simulations. If they do not the discussion of subtle differences between enzyme flexibility and function seems premature. This should be clarified.

We have pointed out on pg. 10 that the systems were chosen due to the availability of X-ray structures. Precisely which structures are used have been listed in the “Molecular dynamics simulations” subsection of the Methods section. Specifically, X-ray structures were available for the *Bacillus licheniformis* (BL), TEM-1, ENCA, GNCA_{MP}, GNCA4 and PNCA lactamases, as well as for the corresponding GNCA4-W229D/F290W and PNCA-W229D variants ((PDB IDs: 4BLM, 1BTL, 3ZDJ, 4B88, 5FQQ, 4C6Y, 4UHU, 5FQI respectively). In the case of the BL-W229D, TEM1-M182T/W229D and PNCA-W229D variants, no crystal structures were available and therefore these substitutions were manually inserted into the wild-type enzyme using PyMOL’s Mutagenesis Wizard. In the case of the TSA complexes, structures were available for both GNCA_{MP}-W229D and GNCA4-W229D/F290W. In the case of the other enzymes, the TSA was manually placed in the active site, using the complex with the GNCA4s as a template (as discussed in the manuscript, the active site shape/dynamical properties do not allow these other enzymes to easily bind the TSA, which is part of the underlying reason for the differences in activity observed for these variants).

¹ Gumulya, Y. & Gillan, E.M.J. Exploring the past and the future of protein evolution with ancestral sequence reconstruction: the “retro” approach to protein engineering. *Biochem. J.* **474**, 1-19 (2017).

At the request of Reviewer 2, we have also extended the simulations to include two further extant β -lactamases along the line of descent leading to the modern Enterobacteria, specifically the β -lactamases from *E. cloacae* and *P. vulgaris*. Crystal structures were available for the wild-type enzymes (PDB IDs 1BUE and 1HZO, respectively), and the W229D substitution/TSA placement in the active site were performed in the same way as for the TEM-1 and BL β -lactamases. The methodology section has been updated on pg. 20 to reflect this fact.

3) The Rex values are quite small ($< 2 \text{ s}^{-1}$). Given the data was only acquired at a single field there are alternative explanations for such small values that do not include us-ms motions. This should be discussed.

The reviewer is right, the exchange contribution was estimated from the line broadening excess using a Lipari-Szabo model free analysis of the relaxation data only at 600 MHz. Even though this data was acquired at a single field, it is important to emphasize that the R2/R1 ratio is very homogeneous, thus indicating that putative deviations from the expected correlation time (due for instance to molecular association events) can be detected accurately. Furthermore, Rex values published by Savard et al.² for TEM-1 β -lactamase using relaxation measurements are exactly in the same range than the ones described here for our GNCA lactamase, in agreement with their similar size and fold. This concordance increases the confidence in our results. Moreover, the rotational diffusion tensor was analyzed using the full relaxation dataset in combination with the crystallographic structure of the protein (PDB 4B88) to rule out that an increased R2 value would be induced by the molecular anisotropy. A sentence providing this explanation has been added in Supplementary Note A, where the detailed results about NMR spectroscopy of the GNCA_{MP} lactamase are described.

Additionally, it is well known that the measurement of relaxation parameters of large molecules at high magnetic fields is not exempt from difficulties. For instance, the expected R1 values for a large protein like GNCA lactamase diminishes as the magnetic fields increases and that NOE values are difficult to measure accurately. For this reason, selection of the appropriated magnetic field to behave the relaxation studies in a large molecule is crucial and defines the accuracy with which these values are determined. We have measured also the relaxation values at 800 MHz, but the calculation convergence was worse than what we had when using only the 600 MHz set. This occurs even in the case of calculations using the procedure described by Ishima et al.³, in which the first points of R1 are discarded for its adjustment to the exponential function. Despite this, it should be noted that in all cases, the global correlation time and the tensor component values were reproducible. For all these reasons, we decided to work with data obtained only at 600 MHz as the addition of that obtained at 800 MHz does not improve the final results.

Interestingly, in the paper of Savard et al.³ for TEM-1 β -lactamase (same size and fold as GNCA), values at 800 MHz were not determined for all the relaxation parameters, the NOE values were not used. This is probably due to the problems mentioned above.

² Savard, P.Y., & Gagné, S.M. Backbone dynamics of TEM-1 determined by NMR: evidence for a highly ordered protein. *Biochemistry* **45**, 11414-11424 (2006).

³ Ishima, R. A probe to monitor performance of N¹⁵ longitudinal relaxation experiments for proteins in solution. *J. Biomol. NMR* **58**, 113-122 (2014).

Likewise, I echo the need for a structure for these analyses. It wasn't immediately obvious from the manuscript.

We have included this information in the revised version “The principal components of the GNCA_{MP} inertia tensor were calculated with the PDBinertia program using the X-ray structure of the GNCA_{MP} β-lactamase (PDB ID: 4B88)” (sentence on page 18). Also, this information is now available in the Figure caption of Figure 3 where the NMR data is shown.

4) minor: DSC is mentioned in the Methods but I couldn't find any figure or results that included DSC.

We obtained DSC data on many of the proteins studied in this work, and this data was included in an early draft of the manuscript. However, these data did not make any substantial contribution to the main point of the manuscript, and were therefore removed due to space constraints, in order to be able to include all the most relevant data. This is therefore a remnant from a previous draft of the manuscript that should have been removed. We thank the Reviewer for catching it, and we have now deleted the paragraph on DSC from the revised version of the manuscript.

Reviewer #2 (Remarks to the Author):

The findings of this study are fascinating and important. The results should definitely be published, and are suitable for Nat Comm. The authors' main discovery that a minimalist change (single amino acid substitution W229D) on ancestral (reconstructed pre-Cambrian) lactamase sequences can lead to significant Kemp elimination activities higher than those achieved by detailed designs (iterative, Rosetta) was unexpected and opens up many new avenues of research into the evolution of protein function and structure.

We thank the Reviewer for their positive assessment of the manuscript, and appreciate that they share our excitement about the results. We have addressed the Reviewer's helpful comments point-by-point below.

There is room for improvement in the study and presentation of this work, however. It would be desirable if the following issues are addressed:

1. In Fig.6, catalytic efficiencies of the present minimalist designed Kemp eliminases are lower by about 2 orders of magnitude than the "best" Kemp eliminase to date from 17 rounds of directed evolution. How feasible is it to use the authors' minimalist sequences as basis for a directed evolution study to ascertain whether the final product of such an evolutionary process can exceed the best Kemp eliminases to date?

This is indeed an exciting idea, as our ancestral scaffolds display highly enhanced stability, a feature that contributes to evolvability. We are in fact currently performing both experimental and computational directed evolution studies for enhanced Kemp elimination activity with the ancestral/engineered lactamases. Our preliminary results are promising, but as they are still quite preliminary (and this is a substantial study in its own right), we felt it a bit premature to speculate whether we will be able to exceed the outstanding Kemp eliminase activity achieved by Hilvert and coworkers. This will rather be the topic of a new independent paper when the directed evolution studies are finished.

2. Experimental atomic structural data were provided for one ancestral sequence in Fig.3 (through NMR) and two ancestral sequences in Fig.4 (through X-ray crystallography), whereas MD simulation data were provided in Figs.8-10 for more sequences. This raises two questions:

(a) Conformational flexibility predicted by MD simulation should be compared directly with experimental structural data, e.g., between B-factors and the stringency of NMR constraints on one hand and the simulated RMSD in Fig.8 on the other hand.

As correctly pointed out by the reviewer, we performed NMR relaxation studies for one ancestral sequence (the corresponding data for the modern TEM-1 protein were available from the literature for comparison). It is to be noted, however, that the main purpose of such a study was to derive information that could be used in the design of the new active site, as we explain in the first section of the Results. Likewise, we performed X-ray crystallography in the presence of a transition-state analog for two ancestral sequences. This was done to obtain experimental evidence of the expected conformational re-arrangements required for the binding of the transition state (*i.e.*, due to the presence of an aspartate residue at the new active site, the transition state must be displaced with respect to the position of the replaced tryptophan residue and the α -helices must shift to allow for such a displacement). Therefore, the NMR experiments of Fig. 3 and the X-ray studies of Fig. 4 were performed to provide specific and essential pieces of information for our work. This been said, we of course acknowledge the Reviewer's point that a clearer comparison between different flexibility metrics would improve the manuscript. Such a comparison is therefore included in the revised version.

Specifically, we point out (see paragraph highlighted in red on pg. 11) that direct agreement between the MD simulations and the residues with exchange contribution to the NMR relaxation (*i.e.*, the information we used to guide active-site design) is not to be expected because of the different time-scales involved. That is, inclusion of exchange terms to explain relaxation rates reveals dynamic processes in the \sim microseconds to \sim milliseconds range, while the timescales typically probed by MD simulations are much shorter. However, our 200 ns MD simulations approach the lower end of the exchange time-scale and, in fact, there is convergence of the two approaches at the region of the new active-site. In order to make this visually clear, we have included in the revised version (Supplementary Fig. 8) a representation of the tertiary structures of TEM-1 and GNCA_{MP} β -lactamases in which both, the region of the *de novo* active site and the residues with exchange contributions are highlighted, showing the congruence of these residues with the regions of increased RMSF at the *de novo* active site in the ancestral protein. For the convenience of the reviewer, this figure is also reproduced below:

Supplementary Figure 8. Tertiary structures of the TEM-1 β -lactamase and GNCA_{MP} β -lactamases with the region of the *de novo* active site and the residues with exchange contributions to the NMR relaxation highlighted. The protein backbone is colored according to the RMSF values calculated from the MD simulations, and it can be seen that the residues with exchange contributions to the NMR relaxation match the elevated RMSF at the *de novo* active-site region. The color scale here is the same as that used in Fig. 9 of the main text. Residues with exchange contributions to the NMR relaxation rates are highlighted by displaying their side-chains in orange.

We also note in the main text of the revised version (pg. 11) that the capability of our MD simulations to capture flexibility features that are relevant for new active-site generation is further supported by the general congruence with the catalytic properties of several modern and ancestral scaffolds.

Regarding the relationship between the results of the MD simulations and the crystallographic B-factors, we note first that, although the variation of B-factors along the polypeptide chain has been often interpreted in dynamical terms⁴, heterogeneity is present in the crystalline form⁵. Furthermore, crystal contacts⁶, the data resolution limit⁷, radiation damage, crystal lattice defects, rigid body motions, occupancy levels and refinement artifacts⁸ can definitively affect the values determined for the B-factors. It is, therefore, debatable whether a correlation between B-factors and the outcome of MD simulations is to be expected on general grounds and the implications of the absence of such a correlation in a given case are far from clear. This being said, for the proteins studied in this work there is qualitative congruence between

⁴ Halle, B. Flexibility and packing in proteins. *Proc. Natl. Acad. Sci. USA* **99**, 1274–1279 (2002).

⁵ De Pristo, M.A., Bakker, P.I.W. & Blundell, T.L. Heterogeneity and inaccuracy in protein structures solved by X-ray crystallography. *Structure* **12**, 831-838 (2004).

⁶ Eyal, E., Gerzon, S., Potapov, V., Edelman, M. & Sobolev, V. The limit of accuracy of protein modeling: influence of crystal packing on protein structure. *J. Mol. Biol.* **351**, 759-771 (2005).

⁷ De Pristo, M.A., Bakker, P.I.W. & Blundell, T.L. Heterogeneity and inaccuracy in protein structures solved by X-ray crystallography. *Structure* **12**, 831-838 (2004).

⁸ Kuzmanic, A., Pannu, N.S. & Zagrovic, B. X-ray refinement significantly underestimates the level of microscopic heterogeneity in biomolecular crystals. *Nat. Commun.* **5**, 3220 (2014).

B-factors and the MD-derived RMSF factors as shown by the profiles of this quantities shown in Supplementary Figs. 18 and 19 included in the Supplementary Information of the revised version. For the reviewer's convenience, these figures are reproduced below. Actually, the congruence observed (regions of high RMSF often, although not always, match regions of high B-factor) appears comparable to that reported by Orozco and co-workers in their consensus analysis of protein dynamics (see Figure 3 in Rueda *et al.* (2007)⁹). In the main text of the revised version, we briefly discuss the potential relationship between the RMSF values and B-factors (pgs. 21-22) and refer the reader to the relevant figures in the Supplementary Information.

Supplementary Figure 18. Root mean square fluctuations and crystallographic B-factors for several modern β -lactamases studied in this work. Shown here are the data for the TEM-1 β -lactamase and for the β -lactamases from *B. licheniformis* (BL), *E. cloacae* (NMC-A) and *P. vulgaris* (Bla-B) are shown. The average residue values for RMSF and B-factor are plotted vs. the residue number. The overall congruence observed (regions of high RMSF often, although not always, match regions of high B-factor) appears comparable to that reported by Orozco and co-workers in their consensus analysis of protein dynamics (see Figure 3 in Rueda *et al.*⁹).

⁹ Rueda, M., Ferrer-Costa, C., Meyer, T., Pérez, A., Camps, J., Hospital, A., Gelpi, J.L. & Orozco, M. A consensus view of protein dynamics. *Proc. Natl. Acad. Sci. USA* **104**, 796-801 (2007).

Supplementary Figure 19. Root mean square fluctuations and crystallographic B-factors for several resurrected Precambrian β -lactamases studied in this work. The average residue values for the RMSF and B-factors are plotted vs. the residue number. The overall congruence observed (regions of high RMSF often, although not always, match regions of high B-factor) appears comparable to that reported by Orozco and co-workers in their consensus analysis of protein dynamics (see Figure 3 in Rueda et al.⁹).

(b) Simulations should also be performed for the extant sequences (those shown on the right of Fig.1) as controls.

Due to the computational cost, it is not possible to perform simulations on all the extant sequences shown on the right of Fig. 1 (these are quite long simulations per system and thus very computationally costly), and we are additionally limited by the availability of high quality structural information. However, we have added two more extant lactamases along the line of descent that leads to the modern Enterobacteria as controls, specifically the lactamases from *E. cloacae* (PDB ID: 1BUE) and *P. vulgaris* (PDB ID: 1HZO). These particular systems were selected both to continue further along the line of descent leading to the modern Enterobacteria, and also because the two selected proteins show comparatively low sequence identity with TEM-1 β -lactamase, thus providing suitable control simulations. We found for

these two additional proteins (Supplementary Fig. 9) the same pattern previously observed for the modern TEM-1 β -lactamase, that is, low flexibility in the region of the *de novo* active site and inability to retain the TSA bound. As we note in the main text of the revised version (pg. 12), this pattern is likely to be a general feature of modern β -lactamases from Enterobacteria. For the convenience of the reviewer, we reproduce Supplementary Fig. 9 below.

Supplementary Figure 9. Results of MD simulations with modern β -lactamases from *E. cloacae* (Nmc-A) and *P. vulgaris* (Bla-B). (A) Tertiary structures of the wild-type and W229D variants colored by RMSF value (see Fig. 9 in the main text for comparison). (B) Distance (in Å) between the nitrogen atom of the 5(6)-nitrobenzotriazole TSA and the closest atom of D229 in simulations of the enzyme-TSA complexes. These simulations are equivalent to those shown in Fig. 10 of the main text. The large values reached for the distance after 100 ns of simulation time indicates that, as with the other modern enzymes studied in this work, the proteins cannot retain the TSA in the pocket generated by the W229D mutation.

Reviewer #3 (Remarks to the Author):

There are several strong points to the manuscript. The combination of MD-simulations, experimental measurements (kM, kcat, NMR), and x-ray crystallography was compelling.

We are delighted that the Reviewer enjoyed our work, and thank them for their helpful suggestions. We have addressed these point-by-point below.

Despite these strengths, there are some points which need further investigation prior for a publication in Nature communication:

1) First of all, I was wondering, how the authors could predict the sequences of the precambrian ancestors so precisely? I couldn't find anything about their sequence analysis in the methods part or supporting information.

The sequence analysis was described in detail in the Supporting Information of the paper we published in JACS in 2013¹⁰, where the resurrected Precambrian β -lactamases were first reported. We took great care to provide as much detail as possible to ensure reproducibility, and a comparison with other studies will illustrate that we have provided far more detail than is typically provided in most published ancestral reconstruction studies. In the revised manuscript, we refer the reader to the Supporting Information of Risso *et al.*¹⁰ (reference 11 of the revised manuscript) for details of the sequence reconstruction (see also the text highlighted in red on pg. 4).

2) Additionally to 1) the authors should implement a short paper research about the basic principle, that ancestral proteins are more promiscuous as modern ones.

We have in fact very recently published a review article in which we deal explicitly (and at length) with the several arguments and recent publications that support the hypothesis that promiscuity should be a common outcome of ancestral protein resurrection¹¹. We hope this work is of interest to the reviewer! We have also explicitly referred to this review in the revised version of the manuscript (see the text highlighted in red on pg. 4).

3) The authors investigated whether the His-tag influences their reaction conditions or not. There is also a Figure of their results in the supplementary. I was wondering, however, why they did not measure kcat/kM, kcat, and kM in the absence of the His tag at pH-levels >8.0. They did so for the protein with His-tag. Could the authors explain, why this is missing?

There is no particular reason. We just did experiments until we were convinced that the His-tag does not have a significant effect. Still, at the reviewer's request, we have determined the profile of rate versus substrate concentration in the absence of His-tag at pH 8.7. The experimental results are shown below.

¹⁰ Risso, V.A., Gavira, J.A., Mejia-Carmona, D., Gaucher, E.A. & Sanchez-Ruiz, J.M. Hyperstability and substrate promiscuity in laboratory resurrections of Precambrian β -lactamases. *J. Am. Chem. Soc.* **135**, 2899-2902 (2013).

¹¹ Risso, V.A. & Sanchez-Ruiz, J.M. (2017). Resurrected ancestral proteins as scaffolds for protein engineering. In: *Directed Enzyme Evolution: Advances and Applications* (Alcalde, M., ed.). Springer International Publishing, pp. 229-255 (2017).

Fit of the Michaelis-Menten equation to the experimental data (continuous line in the plot above) leads to values for the catalytic parameters that are consistent with those previously determined for the His-tagged protein at basic pH. The additional data point at pH 8.7 in the absence of His-tag has been included in the Supplementary Fig. 11 of the revised version. For the reviewer's convenience, the modified figure is shown below.

4) In the discussion of the paper the authors state that “87% of all enzyme functions have evolved from ancestors with generic functionalities and that 13% of enzyme functions have emerged through the generation of new active sites.” Based on this I would expect that the modern lactamases (TEM-1, SABL A, blaF, etc.) would show a stronger Kemp elimination than their ancestors. According to the results presented in this paper, however, this is not true, since the ancestors show a higher Kemp elimination rate than the modern lactamases. If it worked with the ancestors, why should the younger lactamases lose the ability for the Kemp elimination? Is this just because they are not so flexible anymore as their ancestors? If the modern lactamases lost the ability to perform a Kemp elimination than this is, in my opinion, more interesting than that the W229D/F290W alternative of GNCA4 ancestor has a higher Kemp elimination rate than the younger lactamases or the uncatalysed reaction.

What is interesting with the modern lactamases is the fact that unlike the ancestral lactamases, the modern lactamases lose the ability to bind and maintain the TSA (and thus presumably the actual substrate) in a productive conformation long enough for reaction to occur. This can be seen in the distance plots shown in Fig. 10 of the main text, and is also supported by experiment, at least in the case of TEM-1. Specifically, such experiments are now mentioned in the main text of the revised version (pg. 14) and have been included as Supplementary Figure 10. For the Reviewer's convenience, we have appended this figure below:

Supplementary Figure 10. Binding of indole and the transition-state analog 5(6)-nitrobenzotriazole (TS) to ancestral and modern lactamase scaffolds with cavity-creating mutations at position 229. A) Binding to GNCA_{MP}-W229D (left) and GNCA4-W229D/F290W (right) lactamases followed by inhibition of the Kemp elimination. The continuous lines are the best fits of equation 2 in the main text (based on the Michaelis-

Menten mechanism with competitive reversible inhibition). B) Binding to variants of the modern TEM-1 lactamase with the W229D (left) and W229G (right) mutations followed by the rescue of the antibiotic (nitrofecin) degradation activity; in both cases, the global suppressor mutation M182T was included for stabilization. Note that there is no binding of the transition state analog or even indole to the TEM1-W229D/M182T variant. On the other hand, the indole does bind to the W229G variant of the TEM-1 scaffold, although not to the W229D variant (which, in both cases, were previously stabilized by the global suppressor M182T mutation). It appears, therefore, that binding to the modern TEM-1 lactamase is possible, but only when the shape of the ligand matches closely the shape of the cavity generated at position 229 (in a rigid molecular environment, a W229G mutation should generate an indole-shaped cavity, due to the removal of the tryptophan side chain).

When this is combined with the reduced flexibility of the modern enzymes, this suggests that the younger enzymes not only have the wrong architecture to accommodate the *de novo* active site, but also have lost the flexibility to adapt to the TSA once it binds. We agree with the reviewer that it's curious that the enzyme appears to do this!

The Reviewer's comment also made us realize that we failed to convey an essential piece of information about the Kemp elimination reaction in the first version of the manuscript: that is, Kemp elimination is a non-natural reaction, *i.e.* one that is unknown in biological organisms. The 87% vs. 13% statistic refers to *natural* enzyme functions; as Kemp elimination is a non-natural reaction, no enzyme has evolved to catalyze this reaction, and significant Kemp eliminase activity is very rarely found in natural enzymes¹². The notable exception to this is serum albumins, which provide a "special" case, and which we have discussed below in our response to the Reviewer's next comment. Using a non-natural reaction as a model to study the factors that determine the emergence of *de novo* active sites is very convenient, because the interpretation of the experimental results is very unlikely to be compromised by contamination with a natural Kemp eliminase. This is important since contamination by natural enzymes was likely the problem that led to retractions of the first papers on *de novo* enzyme design (in Nature and Science) about 15 years ago. We thank the Reviewer for inadvertently drawing our attention to this point, and have updated the associated discussion on pg. 5 (Introduction), and, in some more detail, on pgs. 7-8 of the revised manuscript.

5) The key aspects for their comparison of the activity of the ancestors and modern lactamases are based on two "model reactions": Kemp elimination and the hydrolysis of an ester, 4-nitrophenyl acetate. These reactions are characterized by a high thermodynamic driving force, low enthalpy of activation, and a single rate-determining transition state (Casey et al., J. Org. Chem., 1973 and Hollfelder et al., letters to Nature, 1996). I would ask the authors to repeat their experiments of the Kemp elimination with BSA and additionally in acetonitrile.

It seems churlish to ask the authors to do another experiment when they have already a lot, but without this crucial experiment, the paper remains a bit of a muddle, and all conclusions tendered must be heavily qualified and couched.

As requested by the reviewer, we have repeated our experiments of the Kemp elimination with BSA and additionally for the catalysis of acetate in acetonitrile. As described below, the

¹² Khersonsky, O., Malitsky, S., Rogachev, I. & Tawfik, D.S. Role of chemistry versus substrate binding in recruiting promiscuous enzyme function. *Biochemistry* **50**, 2683-2690 (2011).

results we have obtained are in excellent agreement with those previously reported in the literature and therefore support the soundness of our experimental protocols.

We have determined the rate of Kemp elimination catalyzed by BSA in the solvent conditions used in most of our experiments: 50 mM HEPES buffer pH 7, 25 °C. The results are given in Supplementary Fig. 12 (see legend to this Figure for experimental details) which is also reproduced below for the Reviewer's convenience. Fitting the Michaelis-Menten equation to these experimental data leads to $k_{\text{cat}}=0.011\pm 0.001\text{ s}^{-1}$ and $K_{\text{m}}=475\pm 90\text{ }\mu\text{M}$. These values are in a good agreement with those reported by Hilvert and coworkers¹³ under slightly different solvent conditions: $k_{\text{cat}}=0.017\pm 0.001\text{ s}^{-1}$ and $K_{\text{m}}=720\pm 68\text{ }\mu\text{M}$ in 10 mM phosphate buffer, 100 mM NaCl pH 7.4, 20 °C.

Supplementary Figure 12. Catalysis of Kemp elimination by bovine serum albumin (BSA). Profiles of rate ($v/[E]_0$) vs. substrate concentration. Experiments were performed with a BSA concentration of $11.5\text{ }\mu\text{M}$ at 25 °C in 50 mM HEPES buffer pH 7. The continuous line is the best fit of the Michaelis-Menten to the experimental data. The catalytic parameters derived from this fit ($k_{\text{cat}}=0.011\pm 0.001\text{ s}^{-1}$ and $K_{\text{m}}=475\pm 90\text{ }\mu\text{M}$) are in good agreement with those reported by Hilvert and coworkers [Kikuchi K, Thorn SN, Hilvert D (1996) Albumin-catalyzed proton transfer. *J Am Chem Soc* 118:8184-8185] under slightly different solvent conditions: $k_{\text{cat}}=0.017\pm 0.001\text{ s}^{-1}$ and $K_{\text{m}}=720\pm 68\text{ }\mu\text{M}$ in 10 mM phosphate buffer, 100 mM NaCl pH 7.4, 20 °C.

We have also determined the rate of Kemp elimination catalyzed by the acetate ion in water/acetonitrile mixture of up to 90% acetonitrile. Our results are in excellent agreement with the values previously reported by Holfelder *et al.*¹⁴, as shown in Supplementary Fig. 13. For the Reviewer's convenience, we reproduce this figure below:

¹³ Kikuchi, K., Thorn, S.N., Hilvert, D. Albumin-catalyzed proton transfer. *J. Am. Chem. Soc.* **118**, 8184-8185 (1996).

¹⁴ Holfelder, F., Kirby, A.J., Tawfik, D.S., Kikuchi, K., & Hilvert, D.S. *et al.* (2000) Characterization of proton-transfer catalysis by serum albumins. *J. Am. Chem. Soc.* **122**, 1022-1029 (2000).

Supplementary Figure 13. Catalysis of Kemp elimination by acetate ion in water/acetonitrile mixtures. The mixed solvents were generated by mixing 100 mM HEPES buffer pH 7 with acetonitrile. Sodium acetate was included and the catalytic constants for the acetate ion were derived from the comparison of the rates obtained in the presence and absence of acetate. Data obtained in this work are shown with red closed data symbols. Open blue symbols represent the values reported by Holfelder et al. [Holfelder F, Kirby AJ, Tawfik DS, Kikuchi K, Hilvert D (2000) *J Am Chem Soc* 122:1022-1029],

We understand that these experiments the reviewer asked us to perform mainly provide an assessment of the reliability of our experimental protocols for the determination of Kemp elimination activity and as such they are briefly referred to in the main text of the revised version (pg. 16). Additionally, as described below, we have included in the revised manuscript discussion on catalysis of Kemp elimination by acetate in acetonitrile and by serum albumins as it sheds light on some important features of the catalysis by engineered/ancestral proteins reported in our work.

As we note in the revised version (pgs. 8-9), it has been known for some time that the catalysis of Kemp elimination by carboxylic acids is subject to a strong medium effect, being strongly accelerated in aprotic solvents. The acetate ion in acetonitrile is in fact an excellent catalyst of Kemp elimination, with a second order rate constant of $2800 \text{ M}^{-1}\cdot\text{s}^{-1}$, a value that has been used as a metric to judge the catalytic efficiency of artificial Kemp eliminases¹⁵. We emphasize that, unlike previous rationally-designed Kemp eliminases (see Fig. 6A), our best eliminase displays a maximum catalytic efficiency ($\sim 5500 \text{ M}^{-1}\cdot\text{s}^{-1}$, Fig. 5A) that exceeds the corresponding acetate in acetonitrile level.

We also discuss the catalysis of Kemp elimination by serum albumins in some detail on pgs. 7-8 of the revised version. We note that serum albumins catalyze, even in the absence of bound metals, a remarkable diversity of non-natural reactions, including the Morita-Baylis-Hillman reaction, the aldol reaction, the Henry reaction, the thio-Michael addition, sulfide oxidation

¹⁵ Korendovych, I.V. & DeGrado, W.F. Catalytic efficiency of designed catalytic proteins. *Curr. Opin. Struct. Biol.* **27**, 113-121 (2014).

and ketone reduction¹⁶. This is due to hydrophobic pockets in the albumin structure, which can bind different substrates (as expected from the biological function of albumins as carriers of a variety of substances), and to the presence of lysine residues with catalytic properties within those pockets. The very special case of serum albumin should not obscure the facts that: 1) Kemp elimination is a non-natural reaction and no enzyme has evolved to be a Kemp eliminase; 2) rational design efforts to engineer artificial Kemp eliminases have met with limited success so far, despite the simplicity of the targeted reaction. We emphasize these two facts on pgs. 7-8.

There were some smaller points:

These are mostly typos. We thank the reviewer for pointing them out and have corrected all of them in the revised version.

1) In the caption of Figure 4 the picture 4E is not mentioned.

This is just a typo. C and D in the last sentence of the legend to the figure should be D and E. It has been corrected in the revised version.

2) On p. 8, l. 224: the authors mentioned something about their ““best” eliminase”. Why is best written like “best”? This implies that it is not the best one.

Actually, we wrote “best” to emphasize that it is our best eliminase, but we realize now that this can be misinterpreted. In the revised version we write best, instead of “best”.

3) On the same page in line 236 there is a t missing after through ...through the irreversible...

4) On p. 10, l. 309 change varinant to variant

Thank you for catching these, they have been corrected in the revised version.

5) On p. 12 l. 349: I appears to me that an “an” is missing: ... reconstruction is an unavoidably ...

The revised version (pg. 14) says “the process of inferring ancestral sequences inherently generates uncertainty”

In summary, I found a lot here to support, but I also thought the paper was missing a) an explanation of the sequence analysis and b) a key experiment the comparison of the Kemp elimination with BSA and the repetition of this reaction in acetonitrile as solvent. Including both would make the paper much stronger. Including both it would certainly merit publication in Nature communication. Without it, it’s a hard call—the paper becomes fundamentally descriptive, notwithstanding all the apparent quantification.

We thank the Reviewer for their overall positive assessment of our work, and hope that the revised version, which includes the requested experiments, is now suitable for publication.

¹⁶ Albanese, D.C.M. & Gaggero, N. Albumin as a promiscuous biocatalyst in organic synthesis. *RSC Adv.* **5**, 10588-10598 (2015).

Reviewer #4 (Remarks to the Author):

The data presented, based on different biochemical approaches, do support the conclusions made by the authors on the relationship between conformational flexibility and directed evolution of enzymes.

I have only one minor comment on the manuscript. Can the author write "beta-lactamase or β -lactamase" instead of "lactamase"?

Of course – this has been updated in the revised manuscript, where we now use β -lactamase throughout.

On the other hands, I have two major comments:

1) Page 8, lines 234-238 and figure 7.

The authors indicates that the replacement of the active site serine by alanine or the inactivation of the beta-lactamase by clavulanic acid do not affect the esterase activity. The figure 7 indicate that the absence of the β -lactamase activity decreases the esterase activity at high substrate concentration. Can the authors discuss this point?

As requested by the reviewer, the point is discussed in some detail in the revised version (pgs. 9 and 10). We meant specifically that the esterase activity at the *de novo* (engineered) active site is not affected by inactivation at the natural active site. There is some low level of catalysis linked to the natural active site. This is to be expected because lactamase hydrolysis and ester hydrolysis are chemically similar. What Fig. 7 shows is that killing the natural active site eliminates its low level of esterase activity, but not, of course, the esterase activity at the *de novo* active site. As we discuss in some detail on pg. 10 of the revised version, our results (Fig. 7) indicate a lower affinity of the ester substrate for the natural active site as compared with that for the *de novo* active site, and a minor natural-site contribution that only becomes apparent when the *de novo* active site is saturated at the higher substrate concentrations.

In addition, although I am convinced that I could not find the procedure in the manuscript, we have no indication on the experimental procedure used to produce the clavulanic acid-enzyme complex.

We apologize for the oversight, and have included in the revised version a brief description of the protocol used to test inhibitor inhibition (the text highlighted in red on pgs. 16-17).

In the case of the class A beta-lactamases, it is clear that clavulanic acid id a potent inactivator. Nevertheless, for different enzymes such as some TEM's variants or the *Bacillus licheniformis* beta-lactamase, the complex is unstable and can lead to its hydrolysis into free enzyme. Did the author check for the stability of the enzyme-clavulanic complex by mass spectrometry?

We did check the stability of the enzyme-clavulanic complex on the basis of the inhibition of the antibiotic degradation activity. As described in some detail in the revised version (paragraph highlighted in red on pgs. 16-17), we actually tested 3 different inhibitors: sulbactam, tazobactam and clavulanic acid. We found the complex of the W229D/F290W

variant of GNCA4 β -lactamase with clavulanic acid to be very stable (but not the complexes with sulbactam and tazobactam), as shown by the fact that the nitrocefin degradation activity was essentially eliminated even after overnight incubation. Specifically, nitrocefin degradation levels after overnight incubation at 4 °C were less than 0.5% those of a non-inhibited sample. For this reason, clavulanic acid was used (after overnight incubation) in the experiment aimed at eliminating the esterase activity linked to the natural/ancestral active site (Fig. 7A).

I would also suggest to the authors to perform a study of the impact of the presence of cefotaxime or benzylpenicillin on the esterase activity.

As suggested by the reviewer, we have performed a study of the presence of the impact of benzylpenicillin on the esterase activity of the W229D/F290W variant of the GNCA4 scaffold. Specifically, benzylpenicillin binding at the natural (antibiotic degradation) active site is expected to eliminate the esterase activity at that site, but not of course the esterase activity at the *de novo* active site. Consequently, the presence of 1 mM penicillin decreases the esterase activity at the higher substrate ester concentrations, but not at the lower substrate ester concentrations. These results (given in Supplementary Fig. 7) are fully consistent with those obtained with the variant modified through the S70A mutation or inactivation with clavulanic acid (Fig. 7A). For the Reviewer's convenience, Supplementary Fig. 7 is reproduced below. Experimental details and further discussion can be found in the legend to the figure.

Supplementary Figure 7. Plot of rate ($v/[E]_0$) vs. substrate (*p*-nitrophenyl acetate) concentration for the esterase activity of the W229D/F290W variant of the ancestral GNCA4 scaffold. Experiments were carried out according to the following protocol: for each substrate concentration, hydrolysis was followed spectrophotometrically for a time sufficient to allow an accurate determination of the rate. Subsequently, 1 mM benzylpenicillin was added and the rate was measured again. This antibiotic concentration is much larger than the K_M value and, therefore, the natural active site is expected to be saturated by the antibiotic despite its hydrolysis. The rates in the absence and presence of benzylpenicillin are shown with closed and open symbols, respectively. The results shown here are in agreement with those obtained by blocking the natural active site by mutation and by inactivation with clavulanic acid (see Fig. 7 in the main text).

2) It is believed that active-site serine β -lactamases are the product of divergent evolution path of the transpeptidase domain of Penicillin Binding Protein. It has been also shown that active site serine β -lactamases can catalyzed the hydrolysis of thioester substrate.

I would suggest to the authors to test their best esterase (GNCA4-W229D/F290W) for its activity against thioester or D-Ala-DAla

This is a very interesting experiment and we do thank the reviewer for suggesting it. We note, however, that the suggested experiment probes catalytic promiscuity at the natural-ancestral active site, while the focus of the submitted paper is the *de novo* active site. That is, in the present work, we were only concerned with the esterase activity at the natural site because we wanted to ascertain that the level of esterase activity at the natural active site is comparatively low and does not compromise our inference of promiscuous esterase activity in the *de novo* active site. Overall, therefore, we feel that while the proposed experiment is clearly a very interesting possibility of course, it does not belong in the submitted work; we will, however, include this experiment in future studies of the natural catalytic promiscuity of the ancestral β -lactamases at their original active site.

Reviewer #1 (Remarks to the Author):

The authors have adequately addressed all of my critiques and concerns.

Reviewer #2 (Remarks to the Author):

The authors have done rather extensive additional work to address the concerns and suggestions raised in my previous report. The additions (3+ new supporting figures) are adequate and represent a significant improvement of the manuscript. As I already stated in the original report, the data are very interesting and should definitely be published. Given the improvement in the presentation of the supporting material, I fully endorse publication of the revised manuscript in Nature Communications.

Reviewer #3 (Remarks to the Author):

Since the authors implemented everything asked into their manuscript, the manuscript (NCOMMS-17-00953A) can now be published in Nature Communications.

Reviewer #4 (Remarks to the Author):

I agree with the additional comments made by the authors.